# Developments for Collagen Hydrolysate in Biological, Biochemical, and Biomedical Domains: A Comprehensive Review

**DOI:** 10.3390/ma14112806

**Published:** 2021-05-25

**Authors:** Muhammad Harris, Johan Potgieter, Kashif Ishfaq, Muhammad Shahzad

**Affiliations:** 1Massey Agrifood (MAF) Digital Labs, Massey University, Palmerston North 4410, New Zealand; m.harris@massey.ac.nz; 2Industrial and Manufacturing Engineering Department, Rachna College of Engineering and Technology, Gujranwala 52250, Pakistan; shahzad27@hotmail.com; 3Industrial and Manufacturing Engineering Department, University of Engineering and Technology, Lahore 54890, Pakistan; kashif.ishfaq@uet.edu.pk

**Keywords:** collagen, hydrolysate, gelatin, processing, hierarchy, species, receptors, disease

## Abstract

The collagen hydrolysate, a proteinic biopeptide, is used for various key functionalities in humans and animals. Numerous reviews explained either individually or a few of following aspects: types, processes, properties, and applications. In the recent developments, various biological, biochemical, and biomedical functionalities are achieved in five aspects: process, type, species, disease, receptors. The receptors are rarely addressed in the past which are an essential stimulus to activate various biomedical and biological activities in the metabolic system of humans and animals. Furthermore, a systematic segregation of the recent developments regarding the five main aspects is not yet reported. This review presents various biological, biochemical, and biomedical functionalities achieved for each of the beforementioned five aspects using a systematic approach. The review proposes a novel three-level hierarchy that aims to associate a specific functionality to a particular aspect and its subcategory. The hierarchy also highlights various key research novelties in a categorical manner that will contribute to future research.

## 1. Introduction

Collagen is one of the known biopolymeric proteins in a human body for various key biological, biochemical, and biomedical functions [1,2]. The proteinic structure is composed of three α (alpha) chains of amino acids (Figure 1a), each with a count of 1014 amino acids constituents. The α (alpha) chains are distinguished with a primary molecular sequence of Gly-X-Y (Glycine-X-Y) [3]. The bioavailability of collagens to the human body entitles the final structure which is constituted of a further fourth stage of biomolecular structures: (1) primary, (2) secondary, (3) tertiary, and (4) quaternary. The primary structure is the basic Gly-X-Y amino sequence with proline and hydroxyproline at X and Y, respectively. The tri-amino acid (Gly-X-Y) formation begins as a “transcript” in rough endoplasmic reticulum (RER) as mRNA for the synthesis of initial polypeptide chains (pre-pro α-chain) [4]. The secondary structure is synthesized by hydrolysis of pre-pro α-chains with enzymes (hydrolases) using oxygen and vitamin C to form hydroxylysine and hydroxyproline in RER lumen. The glycosylation is also a common reaction to induce sugar moieties in pro-α-chain. The tertiary structure is formed through coiling into a left-handed helix of each pro-α-chain at every third residue (Gly). The quaternary structure is formed with the intertwining of three pro-α-chains into a right-handed super helical collagen structure. The hydrogen bond of Gly contents at every third residue and hydroxyl bond of hydroxyproline are responsible for the tight helical conformation [2,4,5,6].

The pro-collagen structure has non-helical residual ends, known as telopeptides, that are categorized with carboxyl (’C) or amino (’N) terminated residues. The count of amino acids residues differs in carboxyl and amino telopeptides depending on the species for collagen extraction [4]. The procollagens are translocated through a process of exocytosis to the extra cellular matrix (ECM), where the telopeptides are cleaved by enzymes (peptidases) to form collagen monomers. The monomers are then gathered in groups proceeded by the reaction with lysyl oxidase to form a stable collagen fibril structure. 

This review includes the literature on collagen hydrolysate or gelatin. The collagen is converted into hydrolysate through extraction from different sources using salt, acid, and enzymes, which cause hydrolysis or proteolysis of the natural peptides [2,4,5,6]. Therefore, the collagen hydrolysate is a hydrolyzed form of denatured collagen [7].

The literature reports a valuable contribution through different reviews on collagen. Liu et al. covers the effects of biological structures on bio-based aspects of different collagen hydrolysate and gelatin [4]. Sionkowska et al. briefly provides the potential applications of collagen hydrolysates in a short review [6]. Liu et al. particularly documented the collagen hydrolysate as a dietary intake for biomedical applications such as osteoarthritis (OA) [8]. Meyer et al. documents one of the most detailed reviews on processing techniques of collagen hydrolysate from different species. The review highlights the effects of thermal and physical processing on collagen hydrolysate properties [5]. Recently, López et al. highlighted the extracted sources, processes, and the potential applications for collagen as a biomaterial with low molecular weight [3]. Another recent effort by Kang et al. provides a unique review on different bioactive peptides in which collagen peptides are presented as a rising option for food- and nutrition-based applications [2]. Kehinde et al. provides a unique recent contribution on antidiabetics-based collagen hydrolysate [1].

The above-mentioned reviews provide a holistic view of either of three domains: (1) processing, (2) effects (physiological, biological, biomedical, physical), and (3) applications. In the light of the abovementioned contributions and recent developments in various domains (processing technologies, source, applications), the hierarchical segregation or breakdown of key research approaches adopted for collagen hydrolysate is yet to be devised. Furthermore, one of the key aspects of receptors or signaling pathways, that are vital neural stimuli to cause biological and biomedical functionalities, is not yet reviewed. This review lists the key research approaches and particular developments for collagen hydrolysate in a unique hierarchy. The purpose of hierarchical breakdown is to provide a simple but elucidated picture of the literature on collagen hydrolysates with the advantage of highlighting the potential unexplored novelties. 

This review presents one of the broadest ranges of research domains for collagen hydrolysate, as shown in Figure 1. To the best of our understanding, based on the literature of the last three decades, the research on collagen hydrolysate can be categorized in five major aspects: (1) processing technologies, (2) targeted receptors, (3) diseases, (4) type of collagen, and (5) species. These five aspects are designated as “level 1” as shown as five vertices of a pentagon in Figure 1. Three of the main five domains (types, species, processes) are further divided into different categories with respect to different forms (kinds) and techniques in level 2. Level 3 further breaks down the three categories (types, hydrostatic, pre-treatment) with respect to functionalities, techniques, disintegration, and modifications. The subsequent discussion will present the research on each of the hierarchical levels in separate sections. Though the breakdown is based on the last three decades, the review is documented with a special concentration on the last three years’ literature.

## 2. Species-Based Literature

The research on species can be divided into two aspects: (1) species as collagen source, (2) species as collagen subject, as shown in Figure 1.

### 2.1. Species as Collagen Source

The species for extraction of collagen/collagen hydrolysate/gelatin are listed in Table 1. The extraction is normally performed from different animal sources such as bovine, porcine, marine, chicken, frog, sheep, lamb, duck, etc. The researchers distinguish their novelty based on collagen hydrolysate extraction from different species and organs. Different organs of the same species are noted with versatile biomedical, biological, environmental, or economic benefits, along with the range of molecular weights as shown in Table 1.

#### 2.1.1. Bovine

Bovine is the one of the oldest animal species for extraction of collagen. El-Sayyad et al. extracts the collagen for bone regeneration from the blood of calves [9]. The histomorphometric analysis shows the bone formation with both protein-free dialysate (calf blood) and DM bone (a bioactive bone grafting material) [9]. Liu et al. [10] reports bovine collagen peptides for inhibiting bone loss. The collagen does not report significant inhibition, but the enhanced trabecular number and separation show prevention of deterioration of bone architecture in the hind limb of rats. Ju et al. reports the comparison of collagen made from bovine tendons through two different methods (acid and pepsin) [11]. The research reports a maximum of ~65% yield [11]. 

Ye et al. reports yak (Bos grunniens) bone collagen peptides for inducing osteogenic properties using 41 metabolites biomarkers [12]. The upward expression level of 20 biomarkers shows the significant capability of yak bone peptide to heal osteoporotic damage. Ye et al. extracts the collagen hydrolysate from yak bone to investigate the activity of osteoblast proliferation [13]. The simulation activity of two peptides out of novel 59 (GPSGPAGKDGRIGQPG, GP-16 and GDRGETGPAGPAGPIGPV, GD-18) shows the osteoblast proliferation due to the hydrogen bonding with the epidermal growth receptor factor (EGRF) based on the manner of dose dependency [13]. 

Zhang et al. reports the collagen extraction from Achilles bovine tendon [14]. The Achilles tendon is destabilized by high pressure and boiling to create an affinity for enzyme hydrolysis. The pre-treatment of bovine collagen followed by enzyme hydrolysis results in improved antihypertensive or angiotensin inhibitory activity (ACE) peptides with good yield [14]. Fu et al. finds a similar kind of ACE inhibitory activity of the bovine nuchals ligament collagen [15]. Lima et al. presents the effect of collagenase and temperature on bovine collagen hydrolysis [16]. The collagenase hydrolysis yields several peptides with <2 KDa, and the increase in temperature decreases the degree of hydrolysis (DH). This shows that the bovine collagen can produce a thermal-dependent yield [16]. O’Sullivan et al. presents the ability of bovine lung collagen hydrolysate to vary antioxidation activity based on hydrolysis by different enzymes [17]. The proinflammatory cytokines interleukin IL-6 and IL-1b show a significant decrease due to an appropriate selection of enzyme (protease) [17].
materials-14-02806-t001_Table 1Table 1Literature regarding species.SourceSource OrganSubjectTypePeptide Sequence/Amino Acid ContentsMWBiological FunctionBovine/calve/yakCalves blood extract [9]RabbitsNot givenNot givenNot givenBone regenerationBone and calcium citrate [10]Tail suspended ovariectomized ratsINot givenNot givenIncreased serum osteocalcin levelsYak bone [12]Ovariectomized ratsINot givenNot givenBone formation biomarkersYak bones collagen [13]Not givenIGPSGPAGKDGRIGQPGGDRGETGPAGPAGPIGPV<3 KDaOsteoblast proliferationBovine Achilles tendon [14]Not givenI44 peptidesGERGFPGLPGPS29 peptides with <1 KDaAngiotensin-converting enzyme (ACE)-inhibitory activityBovine Achilles tendon [16]Not givenI(1) Gly-Asp-Hyl-Gly-Glu-Thr-Gly-Glu-Gln-Gly-Asp-Arg,(2) Phe-Leu-Pro-Gln-Pro-Pro-Gln-Glu-Lys-Ala-His-Asp-Gly-Gly-ArgSeveral peptides <2 KDaAntimicrobial and radical scavengingBovine lung tissue [17]Not givenNot clearNot givenNot givenanti-inflammatoryBovine Nuchal ligament [15]Not givenIGly-Pro-Arg-Gly-Phe (not ACE inhibited)Tyr-Trp and Leu-Arg-Tyr (ACE Inhibited)Not givenACE inhibitor peptidesPorcinePorcine skin [18]Mouse lymphoma L5178YTK ±(not a mouse)Majorly IPeptides of ii, iii and ivNot given<K3DaChronic wound healingporcine leg bones [19]Osteoblast MC3T3-E1 cells (not a mouse)INot given1 to >5 KDaOvercome osteoporosis effects on bonesFresh pig bone [20]Osteoblast MC3T3-E1 cells (not a mouse)INot givenNot givenUpregulating osteoblastsPig skin gelatin water extracts [21]Male ICR miceNot givenNot given<3 KDaCure of cognitivefunctionPorcine gelatin [22]HumanNot givenNot given1200 Dalanguage cognitive function and brain structurePig skin gelatin [23]Human neuroblastoma cell line (SH-SY5Y)Not givenNot given>50 KDaAntioxidative and neuroprotective effectMarinePacific oyster (Crassostrea gigas) [24]SKH hairless mice (Mice dorsal skins)IIle-Val-Val-Pro-Lys554.72 DaAnti-photoagingFresh water Salmo Salar skinDeep sea Tilapia Nilotica skin [25]RatNot givenAsp, Glu, Ser, His, Gly, Thr, Arg, Ala, Tyr, Val, Met, Phe, Ile, Leu, Lys, Pro, Hyp512.63 Da for Tilapia NiloticaAltering Cutaneous MicrobiomeColonization for wound healingRedlip Croaker (Pseudosciaena polyactis) Scales [26]Not givenIDGPEGR,GPEGPMGLE, EGPFGPEG, YGPDGPTG, GFIGPTE, and IGPLGA629.6, 885.9, 788.9, 762.7, 733.8, and 526.6 DaAnti-oxidation scavengingSoft shell of turtle (Carapax Trionycis) [27]SD ratsNot givenAmino acid w.r.t mg/100 g Ser > Gly > Tyr > Ala > Glu > Asp > Cys > Val> Arg > Lys > Thr > Ile > Met > Leu > Try > His > PheNot givenPre-biotic potentialSwim bladders of Atlantic cod (Gadus morhua) [28]Human fibro blast cell (MRC-5)INot givenACS 182.9 ± 7.9PSC28.7 ± 1.7 KDaHigh thermal propertiesMackerel fillets (Scomber scombrus) [29]Not givenNot givenAspartic acid (Asp), Serine (Ser), Glutamic acid (Glu), Glycine (Gly), Arginine (Arg), Threonine (Thr), Alanine (Ala), Proline (Pro), Tyrosine (Tyr), Valine (Val), Methionine (Met), Lysine (Lys), Isoleucine (Ile), Leucine (Leu), Phenylalanine (Phe)Not givenAntioxidation and antihypertensionAsian swamp eels [30]Not givenNot givenAsp, Glu, Lys, His, Arg, Asn, Gln, Ser and ThrNot givenEmulsifier and stabilizer in food productsSharks [31]MiceNot givenAsn, Thr, Ser, Glu, Pro, Gly, Ala, Val, Cys, Met, Leu,Ile, Tyr, Phe, His, Lys, Arg, Hyp.<800 DaOsteoblast proliferationChickenSternal cartilage [32]Not givenIIGlu, Ala, Pro, and Hyp,Cys, Met, Ile, Tyr, Phe, His245 KDaType II collagenChicken Heads (gelatin) [33]Not givenNot givenNot givenNot givenGood use of by-productSkin (gelatin) [34]Not givenBHydroxyprolineNot givenGood use of by-productFrogRana chensinensis skin [35]Not givenNot givenAla, Arg, Asp, Cys, Glu, Gly, His, Iso, Leu, Lys, Met, Phe, Hyp, Pro, Ser, Thr Tyr, Val106 KDaFood stuff and medicalDuckBreast [36]Not givenNot givenAsp, Thr, Ser, Glu, Gly, Ala, Cys, Val, Met, Ile, Leu, Tyr, Phe, Lys, His, Arg, Pro491–715 DaAntioxidantSheep and lambsBone, cartilage, carcass trimmings and meat [37]Not givenINot given5 < MW < 100 KDaGood use of by-productBeesRoyal Jelly [38]DrosophilaNot givenGly, Glu, asp, Arg, Pro, Leu, Ala and Val<1 KDaAnti-oxidative, anti-aging

#### 2.1.2. Porcine

The recent research work on porcine collagen/collagen hydrolysates/gelatin is shown in Table 1. Wang et al. presents the potential porcine skin collagen matrix for human wound healing extracted through supercritical carbon dioxide treatment [19]. The matrix shows high biocompatibility and non-toxicity with wound healing [19]. Zhu et al. prepared two different molecular weight collagen peptides with porcine leg bone for the prevention of osteoporosis activity [19]. The mouse osteoblast MC3T3-E1 cells show significant proliferation activity for low molecular weight (<1 KDa). Furthermore, the collagen peptides cause a decrease in Bax/Bcl-2 expression to inhibit the apoptosis [19]. Wu et al. reports a similar research on the proliferation of osteoblast MC3T3-E1 cells through the use of collagen-extracted peptides of fresh pig bone [20]. Kim et al. extracts the low molecular weight gelatin from pig skin to repair cognitive functions [21]. The low molecular weight hydrolysate in a high dose (400 mg/kg) attenuates the acetylcholine (ACh) esterase activity and significantly enhances the ACh contents that cause an increase in antioxidative properties [21]. Koizumi et al. also reports improvement in cognitive functions related to language due to the 5 g oral ingestion of porcine gelatin [22]. Another recent research by Bechaux et al. reports the collagen hydrolysate of heart, liver, and lung and a muscle (Longissimus Dorsi) of porcine [39]. The authors designate the extracted peptides as anti-metabolic syndrome peptides. These particular biopeptides result in high bioactive functionalities [39]. 

#### 2.1.3. Marine

Bang et al. proposes pacific oyster hydrolysate for anti-photoaging applications [24]. The hydrolysate provides novel pentapeptides along with other useful peptides to cause anti-inflammatory effects on SKH-1 hairless mice [24]. Mei et al. [25] prepares the bio-peptides from salmo salar skin (Ss-SCPs) and tilapia nilotica skin (Tn-SCPs) for wound healing, wound microflora colonization, and collagen formation in an extra cellular matrix (ECM). The wound shows angiogenesis, an increase in collagen deposition, and a reduction in inflammation due to the up-regulation of antimicrobial peptides (β-defence14, BD14) and a pattern recognition receptor (NOD2) [25]. Wang et al. [27] provides the significant change in the degree of hydrolysis and radical scavenging with 2,2-diphenyl-1-picrylhydrazyl (DPPH.) of red-lip croaker scales hydrolyzation with six different proteases. Six peptides sequences show protection of HepG2 cells from oxidative damage (H_2_O_2_) through: (a) an activation of the antioxidative enzymes (superoxide dismutase, catalase, and glutathione peroxidase), and (b) a decrease in reactive oxygen species (ROS) [27]. Korczek et al. proposes a novel hydrolysate of Atlantic mackerel with good thermal endurance as a pre-treatment combined with enzymatic hydrolysis [29]. The hydrolysate is found suitable as a functional food ingredient due to significant DPPH activity along with iron (Fe^2+^)-chelating ability [29]. Wang et al. proposes a novel vinegar quenching method along with super fine grinding to make soft shell turtle bioactive peptides [27]. The proven calcium-based bio-availability makes the super fine powder suitable for food supplements [27]. Sousa et al. performs hydrolysis to make a novel hydrolysate of the swim bladders of Atlantic cod [28]. Along with the proper yield of collagen peptides from a novel source, the human cell culturing with novel collagen hydrolysate shows no cytotoxicity, making it a feasible choice for different food and health applications [28]. Halim et al. presents the novel Asian swamp eel hydrolysate as a food stabilizer and emulsifier with good emulsification and foaming properties [30]. Xu et al. uses two types of sharks to make three novel collagen peptides for osteoblast activity [31]. The resulting collagen peptides reveal a molecular weight of <800 Da with high mineral bone density and osteoblast proliferation [31].

#### 2.1.4. Chicken

Akram et al. prepares the chicken sternal cartilage hydrolysate with ultrasonication [32]. The extracted hydrolysate has a good yield. Gál et al. reports the chicken head gelatin to have good gelation properties such as gel strength, viscosity, and melting temperature [33]. The gel strength, viscosity, and melting temperature are noted with a maximum of 335 bloom, 9.5 mPa.s, and 42 °C, respectively [33]. Bichukale et al. investigates the poultry skin gelatin for valorization of poultry waste [34]. The temperature-based effects during preparation are found to be effective to vary the yield and gelatin properties [34]. Recently, Alves et al. registered the first research on collagen hydrolysate from chicken blood meal to increase the utilization of slaughter by-products [40]. The hydrolysate with a MW < 14.4 KDa shows high thermal stability with good antioxidative properties found in DPPH and ABTS analysis [40].

All of these recent efforts aim to make use of poultry-based waste; therefore, the effects of recently developed chicken hydrolysates/gelatin from sternal cartilage [32], head [33], and skin [34] are not investigated for any particular biomedical purpose, for example, antioxidation, wound healing, cognitive functions. Therefore, this reveals a novel area to explore in the future.

#### 2.1.5. Others Recent Sources

Zhao et al. extracts the collagen hydrolysate from the skins of frogs (Rana chensinensis) [35]. The proposed hydrolysate has a yield of 15.1% (*w*/*w*) with a high content of glycine (204.5 out of 1000 residues) that is highly suitable for nutritional intake [35]. Vidal et al. proposes the extraction of collagen from lamb and sheep slaughter by-products [37]. The lamb has a comparatively higher yield of 18% (based on dry weight). The sheep collagen has more viscosity and emulsification power (69.6 m^2^/g) [37]. The research reports a wide range of molecular weights for detected peptides (5 to 100 KDa) [37]. Li et al. isolates 11 novel antioxidant peptides from duck breast protein hydrolysate. These novel peptides are all low molecular weight (<1 KDa), and are capable of DPPH inhibition activity along with ABBTS. + scavenging. Qui et al. extracts the enzyme-treated royal jelly (ERJ) collagen for investigating the anti-aging and anti-oxidation effects on drosophila [38]. The intake of 3 mg/mL of ERJ results in an increase of life span of drosophila by 11.16% through reducing protein carbonyl (PCO) and malondialdehyde (MDA) levels [38]. Furthermore, the upregulation of glutathione peroxidase (GSH-Px), total super oxide dismutase (TSOD), and catalase (CAT) results in high anti-oxidation [38]. 

### 2.2. Species as Collagen Hydrolysate Subjects

Different species are used for analyzing the effects of collagen/collagen hydrolysate products as shown in Table 1. These species (subjects) either suffer any corresponding impairment or they are induced with any relevant impairment. A major part of the literature reports rats or mice as a test subject. However, there are other animal subjects (such as rabbits, dogs, cats) treated in pre-clinical treatments with different collagens or collagen peptides or collagen hydrolysate or collagen films. Human patients are also reported in various collagen-based studies as a clinical subject.

Liu et al. investigates the tail-suspended ovariectomized rats for oral ingestion (gastric gavage) to achieve bone regeneration [10]. Ye et al. uses ovariectomized rats for intragastric ingestion of collagen peptides [12]. Kim et al. experiments with ICR mice for the oral ingestion of the novel pig gelatin to achieve improvement in cognitive functions [21]. Bang et al. observes the ultraviolet irradiation inflammatory effects on SKH hair-less mice [24]. Mei et al. analyzes the effects of salmon salar and tilapia nilotica skin collagen peptides on a wounded rat [25]. Most of the osteoarthritis (OA) studies are conducted on rats/mice. Nogimura et al. finds the antidepressant effects of collagen dipeptide (prolyl-hydroxyproline) in male ddY mice [41]. Sonklin et al. also performs an antihypertensive analysis on spontaneously hypertensive rats [36]. Xu et al. uses the C57BL/6 mice for oral ingestion of three novel compound peptides from sharks [31]. Another researcher also reports mice as subjects for analysis of skin impairments [42].

Araújo et el. studies the effects of collagen membrane in a dog’s bone formation [43]. Bornstein et al. reveals the bone regeneration in a dog’s mandibles with the bio-absorbable collagen membranes [44]. Minami et al. considers the canines for administration of effects of collagen hydrolysate [45].

Tal et al. reports a healing process in cats using tissue engineering [46]. The authors provide the histological studies of collagen membranes (cross-linked and non-cross-linked) for cats impaired with perforations in soft tissues and raised mini flaps [46]. Minami et al. publishes one of the few research projects on felines for collagen ingestion effects [45]. Zentek et al. applies the deep analysis of urinary excretion of felines after ingestion of collagen hydrolysate [47]. 

Koizumi et al. performs the language-based cognitive experimentation on humans from 49 to 63 years [22]. Kimura et al. studies the improvement in vocal fold paralysis in human patients after collagen ingestion [48]. Hays et al. reports the effects of whey protein hydrolysate on different aspects (body composition and nitrogen balance) of women patients [49].

El-Sayyad et al. conducted one of the rare studies that uses rabbits for the recovery of calvarial bone [9].

This concludes that analyses of collagen hydrolysate on rabbits, cats, and dogs are not extensively performed, which is a big research area to explore.

## 3. Collagen Type

One of the approaches for research is related to the “Type” of collagen. To the best of our knowledge [50,51,52,53], most of the research work is oriented with either one or two of the following categories: (1) use of particular Type collagen hydrolysate aimed to improve/solve any biomedical impairment, (2) extraction of a particular Type of collagen/collagen product using any novel method (e.g., ultraviolet irradiations, UV), (3) extraction of a particular Type of collagen from any novel source, and (4) bio-synthesis or culturing of modified collagen to enhance different properties. The segregation is shown in Figure 2 and described in the following section. 

### 3.1. Type I

The research of Type I collagen is associated with four categories (Table 2) as given below.

#### 3.1.1. Aimed Biomedical Function for Type I

Watanabe-Kamiyama et al. reports one of the earliest studies on oral ingestion of two types of radioactive Type I hydrolysate (proline, Glycine-proline-hydroxyproline) for osteoporosis in ovary-extracted (OVX) stroke-prone spontaneously hypertensive rats (SHRSPs) [54]. The extraction is performed on chicken feet using proteases hydrolysis. The study holds novelty for the low molecular weight hydrolysate due to the number of bio-chemical analyses (blood, liver, kidney, spleen, cartilage, femurs, tibias, brain, skeletal muscle, and skin). The tri-peptides remained in plasma for 14 days with final accumulation in a kidney. The organic substances are increased, and water contents are decreased in the left femur causing a decrease in osteoporosis, hence resulting in tissue engineering [54]. Recently, Sangsuwan et al. investigated the effects on photoaging in human skin due to the oral ingestion of Type I collagen hydrolysate [55]. Skin elasticity has shown remarkable enhancement in women aged between 50 to 60 suffering from connective skin tissue disorders [55]. Further details are provided in Table 2.

#### 3.1.2. Novel Method for Type I

Abdollahi et al. introduces a unique pH shift process for separating gutted silver fish sediments (skin, bones, scale, connective tissues) to extract Type I collagen and collagen hydrolysate [56] as shown in Figure 3. The sediments-based extraction provides better yield as compared to acid solubilized collagen (ASC). Furthermore, a novel implementation of sequential enzymatic hydrolysis (pepsin then trypsin) leads to high molecular weight collagen hydrolysate that shows high gelation strength when solubilized in proteins of the same gutted silver carp sediments [56]. The extraction from sediments has not yet been reported for other sources such as bovine, lamb, goat, etc., which can be explored in future research.

#### 3.1.3. Novel Source for Type I

Nagai et al. synthesized a novel emu skin Type I hydrolysate through enzymatic hydrolysis using six enzymes (Pepsin, trypsin, papain, chymotrypsin, pronase E, thermolysine) [57]. The novel hydrolysate reveals highest antioxidation for pepsin and chymotrypsin. Tage et al. reports the novel ginger proteases for hydrolyzing the Type I bovine collagen [58]. The novel hydrolysate is found rich in X-Hype-Gly tripeptides that show an increase in their degree of mineralization to achieve high differentiation for mouse MC3T3-E1 pre-osteoblasts [58]. Recently, Zhang et al. proposed a novel deer sinew collagen hydrolysate for dermal (skin) use [59]. The percutaneous proteins from novel collagen induce significant cell proliferation of NIH3T3 cells. Furthermore, the 2,2′-azinobis-3-ethylbenzothiazoline-6-sulfonic acid (ABTS) assay forms a cation that shows ~70% antioxidant activity of percutaneous proteins due to high contents of Gly, Ala, Pro, and Leu [59]. Another recent research is reported by Bang et al. on novel pacific oyster hydrolysate to inhibit and cure skin photoaging due to ultraviolet radiation [24]. The novel peptides purified from oyster hydrolysate show an inhabitation of matrix metalloproteinases (MMPs) and stimulation of the TGFβ/Smad signaling pathway to increase the Type I collagen expressions. The authors also noted a reduction in wrinkle depth, length, and epidermal thickness [24]. The research reports no effect on collagen Type III contents. Therefore, based on the cultured hydrolysate in [60], the Type III collagen alongside Type I can be artificially induced into deer sinew collagen hydrolysate for future research.

#### 3.1.4. Biosynthesis and Culturing for Type I

Recently, Meng et al. performed the fibrogenesis of Type I collagen from swim bladders (SB) and skin with cultured phosphate ions (Pi) [61]. The fibril morphology and thermal properties are regulated with optimal concentration of phosphate ions. The increase in Pi leads to a decrease in the rate of fibril formation and an increase in fibril thickness. The research proposes the SB fibrils for scaffolds due to the diverse morphologies and good thermal characteristics [61]. Another recent work by Liu et al. presents the lipid accumulation to proliferate the preadipocytes (3T3-L1) for development of adipose tissues [62]. The proliferation of preadipocytes is increased with the downregulation of Yes associated proteins (YAP) found in a cultured Type 1 collagen assay [62]. A similar nature of research is reported by the same author, but with a different research group [63], about the effects of collagen Type I on cell migration and differentiation through the culturing of mouse C2C12 myoblasts for skeletal muscle regeneration. The collagen I increases the production of interleukin-6 using nuclear factor κB (NF-κB) p65 that results in the myogenic differentiation of C2C12 cells in the cultured assay [63].
materials-14-02806-t002_Table 2Table 2Examples for Type I collagen or collagen hydrolysate.SourceCategoryMethodDaysPurposeResultsChicken-foot collagen hydrolysate [53]Novel Source and Biomedical functionProteases (enzymatic hydrolysis)
To explain the mechanism for absorbance of low molecular weight hydrolysateOsteoporosis is decreased and the following are increased, Organic substance, tri-peptides in plasmaEmu skin [57]Novel sourceEnzymatic hydrolysis (pepsin, trypsin, papain, chymotrypsin, pronase E, thermolysine)
To make Type collagen hydrolysate with Emu skinBetter antioxidant and anti-allergic propertiesGutted silver carp [56]Novel Method and sourcepH shift for extracting pure sediments only, then sequential pepsin trypsin hydrolysis
Exploring the potential of sediments from pH shift for collagen/collagen hydrolysateHigh molecular weight collagen with better gel strengthBovine bone [58]Novel source and partially BiosynthesisNovel Ginger proteases (papain family)MC3T3-E1 pre-osteoblasts culturing for 7 daysExplore the potential of novel proteases for osteoporosisNovel X-Hype-Gly,1.9-fold more MC3T3-E1 mineralizationDeer sinew [59]Novel SourceSequential Enzyme hydrolysis (pepsin then trypsin)
A novel dermal collagen hydrolysateOxidation decreased, Cell proliferation of NIH3T3 fibroblasts increasesPatent (unknown) [55]Biomedical function Not given4 weeksEffect of collagen hydrolysate in skins under sun-exposed areasElasticity in skin was enhanced

### 3.2. Type II

The research of Type II collagen is associated with four categories as discussed below.

#### 3.2.1. Aimed Biomedical Function for Type II

Osser et al. reports the accumulation in cartilage for attenuating osteoporosis through proliferated stimulation of collagen Type II [64]. The hydrolysate is observed to pass the mucosal barrier without enzymatic cleavage that causes enhanced differentiation [64,65,66]. Ng et al. reports the healing of articular cartilage using tissue engineering. The authors incubate the hydrolysate with 3-dimensional chondrocytes to enhance the efficacy of hydrolysate for Type II collagen application [67]. The previous studies reported in this section only consider the effects on non-athletic patients suffering from osteoarthritic pain. Clark et al. presents the significant reduction of an athlete’s osteoarthritis symptoms (joint pain) after an oral ingestion of 10 g/day of collagen hydrolysate for 24 weeks [50]. This study is one of the earliest research projects to report a control group for clinical trials along with a placebo and a treated group in the collagen Type II literature [50]. Lugo et al. presents 180 days of oral administration of collagen Type II (denatured) hydrolysate extracted from chicken sternum cartilage for decreasing knee osteoarthritis (OA) pain as compared to two groups: (a) glucosamine hydrochloride/chondroitin sulfate (GC), and (b) placebo [68]. The Western Ontario McMaster (WOMAC) score shows the high tolerability and pain relief in collagen Type II participants as compared to the other two testing counterparts [68]. Maneeli et al. published one of the few research projects on low dosage native collagen Type II of porcine for curing the unilateral osteoarthritis in rats caused by sodium mono iodoacetate (MIA) injection (injected interarticular) [69]. The outcomes present protection of cartilage through a decrease in levels of C-terminal cross-linked telopeptide of Type II collagen (CTX-II). The decrease in levels of CTX-II in plasma and urine along with an improvement in postural balance and better motor motion activity are the key performance indicators for Type II native collagen [69]. Table 3 enlists the Type II collagen-aimed biomedical functions.

#### 3.2.2. Novel Method for Type II

Recently, Senra et al. reports a novel manufacturing process of compression and extrusion molding for the combination of ultra-high molecular weight polyethylene (UHMWPE) with two types of collagen (hydrolyzed and Type II) [70]. The blend prepared by twin screw extrusion shows better dispersion of collagen in UHMWPE with an unaffected high temperature of degradation in thermogravimetric analysis (300 °C for collagen and 480 °C for UHMWPW). The hydrolyzed collagen provides better extrusion as compared to Type II owing to the smaller molecular weight at low torques [70]. 

Jeevithan et al. isolates the Type II collagen hydrolysate from a whale shark using pepsin soluble (enzymatic) and acid soluble techniques [71]. The structural similarities of both collagens (pepsin and acid) lead to a strong tolerance by a secondary peptide (helical) structure to acid hydrolysis. However, pepsin collagen provides high protein content, hydroxyproline content, and denaturation temperature as compared to acid soluble collagen (ASC) [71].
materials-14-02806-t003_Table 3Table 3Examples for Type II collagen or collagen hydrolysate.SourceCategoryMethodDaysPurposeResultsNot given [50]Biomedical functionNot given24 weeksOA joint pain symptoms after hydrolysate ingestion for 24 weeksOA athlete’s pain is significantly decreasedArticular bovine chondrocytes, collagen hydrolysate source is not given [60]Bio-synthesis and culturingArticular cartilage chondrocytes by enzymatic hydrolysis using pronase and collagenase. Collagen hydrolysate source is not given.14, 28, 42 daysCulturing of hydrolysate in chondrocyte and agarose gel to enhance Type II collagen in engineered Compressive properties, Type II collagen are improved.Whale shark (WS) cartilage [71]Novel Method and SourceEnzymatic (pepsin) and acid hydrolysis.
Whale shark is not extracted before with acid and pepsin hydrolysis.High denaturation temperature, hydroxyproline, and protein content of pepsin soluble method as compared to acid hydrolysis.Chicken sternum cartilage extract [72]Novel SourceNot given 14 weeksUsing collagen II to reduce skin problems.Reduction of wrinkles, inhibiting cell damage due to UV rays.Chicken sternum cartilage [68]Biomedical functionNot given 180 days

Type II from bovine cartilage [70]Novel methodAcidic hydrolysis
A novel technique and material for orthopedic problems.Better dispersion, mechanical properties.Porcine [69]Biomedical functionNot given13 daysLow dosage effects of Type II on rat OA.Less CTX-II,Better postural balance,Better motor activity.

#### 3.2.3. Novel Source for Type II

Recently, Phipps et al. reported the oral ingestion of collagen Type II extracts by mice for 14 weeks to enhance the ultraviolet-based photoaging in skin [72]. It is the only extract from a single source (chicken cartilage sternum) in the literature that is comprised of collagen Type II peptides, chondroitin sulfate, and hyaluronic acid. The results reveal a significant reduction in wrinkles, matrix metalloproteinase expression (MMP1 and MMP2), dermal inflammatory cells, and trans-epidermal water loss. Furthermore, the hyaluronic acid contents (+24%) and skin elasticity are increased significantly as compared to control UV-irradiated mice [72]. This research highlights the need to explore novel sources with multiple types of proteins for other diseases such as osteoarthritis, dysplasia, etc.

#### 3.2.4. Biosynthesis and Culturing for Type II

Ng et al. performs in-lab bio-synthesis to culture the chondrocytes with agarose and hydrogel for a longer number of days as compared to past studies (42 days) [60]. They found the proliferation in Type II collagen and improved mechanical properties (compressive) [60]. A similar kind of study is extended to find the hydrolysate concentration and the quantity of Type II collagen based on a variable dose intake (grams) [65]. The hydrolysate is incubated in chondrocytes cultures before clinical trials to analyze the effects on accumulation of peptides in cartilage for the biomedical remedy of osteoarthritis (OA) [65].

### 3.3. Type III

There is a vast literature presented on sources [3] and different phenomena associated with Type III collagen. However, unlike Types I and II, the Type III hydrolysate-based literature is rare. It is mostly reported as a combined outcome along with Type I and Type IV in the publications from any particular source [73]. To the best of our knowledge, the literature on Type III collagen hydrolysate falls only into two categories (aimed biomedical function and novel source), as explained in the following discussion. 

#### 3.3.1. Aimed Biomedical Function for Type III

Wang et al. reports the investigation on skin chronological aging of mice using collagen hydrolysate from the Nile tilapia scale [74]. The histochemical staining reveals the dense collagen fibers with an improved Type I/Type III ratio. The antioxidant enzymes (superoxide dismutase and glutathione peroxidase) expressions and collagen contents show statistically significant increases [74]. A similar kind of research with more detailed analysis is reported by Song et al. for chronological-aged mice, and reports the effects of alcalase and collagenase-based bovine collagen peptides [75]. The alcalase provides better results for normalizing the Type I/Type III ratio along with improved skin laxity, collagen fibers, and collagen contents [75]. Kashiuchi et al. presents the histological and physiological effects of oral administration of chicken cartilage hydrolysate, chondroitin sulfate, and glucosamine on rheumatoid arthritis in rats [76]. The relative expression of the Type III alpha chain is decreased more for combined ingestion of collagen hydrolysate and glucosamine [76]. The research highlights a unique and opposite biological behavior of glucosamine in the form of a decrease in collagen I expression, and the corresponding primary oligonucleotide sequences of chsy1 (CAAGTGTCTCCGGGAAATGTCTGGTACGGTGGGTTTTTGT) and chpf (CAACGACATCGTCAGTGCTC AAAAGCTTTGTGCAGCTGGT) [76]. Generally, glucosamine is known for an increase in collagen expression. This highlights a need to conduct independent research on the unusual negative effects of glucosamine. 

#### 3.3.2. Novel Sources for Type III

Bolke et al. finds the 13% of Type III alpha 1 (α1) chains along with 31% of Type I alpha 1 (α1) chains and 18% of Type I alpha 2 (α2) chains in the amino acid sequence obtained from bovine collagen in liquid chromatography–mass spectrometry (LC-MS) [77]. The dose-dependent intake of a food-derived (walnut protein) hydrolysate reports an insignificant increase of Type III collagen and a significant increase in hydroxyproline (Hyp), hyaluronic acid (HA), and Type I [73]. The inhibition of matrix metalloproteinase (MMP-1) activity and relief of epidermal hyperplasia results in the maintenance of skin damaged due to ultraviolet irradiation [73]. Fan et al. presents a novel collagen and collagen hydrolysate of a jellyfish umbrella for the anti-photoaging of mice skin [78]. The collagen hydrolysate shows better antiphotoaging properties as compared to collagen. Other benefits reported for collagen hydrolysate are: maintained ratio of Type I/Type III collagen, better moisture retention, repair of elastin protein fiber, and endogenous collagen of mice skin [78]. Wang et al. reports the novel gelatin hydrolysate of Amur sturgeon (swim bladder) for anti-skin aging effects [79]. The oral ingestion in rats shows the highest absorption in jejunum during simulated gastrointestinal fluid digestion. The ratio of Type III/Type I is significantly decreased (~44%) [79]. The research is unique as it does not report a high concentration in the kidney of the hydrolysate. The high concentration in the kidney is linked with impaired urinary secretion in cats that may end with a severe medical dysfunction.

## 4. Process

There are two conventional methods to synthesize collagen hydrolysate: (1) chemical hydrolysis, and (2) enzymatic hydrolysis [80,81]. The recent literature reports the combination of both chemical and enzymatic hydrolysis for extraction as shown in Figure 4. Apart from these two basic methods, there are various different types of techniques to extract collagen hydrolysate [82]. These technologies are applied as: (a) pre-treatment, i.e., before processing (hydrolyzing) the collagen source, (b) the main extraction process, (c) post-treatment, i.e., after processing (hydrolyzing). This section provides the details of the two conventional and recently developed techniques to extract collagen or collagen hydrolysate or gelatin. 

Figure 4 describes various steps gathered from a wide set of literatures [3,83,84,85,86,87,88,89,90,91,92,93,94,95,96,97,98,99]. The diagonal in Figure 4 shows the sequence of processes for producing the collagen hydrolysate though combined chemical and enzymatic hydrolysis. The lower left triangular section to the diagonal shows the processing parameters for individual processes, and the upper right triangular section presents the categorical segregation of particular individual processes mentioned in the diagonal. For example, the pretreatment is performed using four parameters: (1) incubation time, (2) incubation temperature, (3) source/solvent, and (4) concentration. Similarly, the enzymatic hydrolysis is performed using four parameters: (a) pre-enzyme hydrolysis incubation, (b) enzyme to substrate ratio, (c) time for hydrolysis, (d) concentration. Drying is performed using two parameters: (1) time, and (2) temperature. Filtration is performed using one parameter of “pore size”. In the upper right section, there are found three types of pretreatments: (a) alkaline, (b) acidic, and (c) ultrasound. Similarly, two types of enzymatic hydrolysis are noted: (a) single enzyme, and (b) sequential dual enzymes. The filtration is also found in two types: (a) vacuum, and (b) non-vacuum.

### 4.1. Chemical Hydrolysis

In chemical hydrolysis (acid hydrolysis), the literature reports various organic acids for acid hydrolysis, e.g., acetic acid, citric acid, and lactic acid. Inorganic acid such as hydrochloric acid (HCL) is also reported. However, the efficacy is lower than for organic acids. Acetic acid is the most common with high efficacy and yield [3,98,99]. The collagen extracted using acid hydrolysis is also known as acid soluble collagens (ASC). The concentration, incubation temperature, incubation time, and ratio of raw material to acid solution are important parameters for controlling the optimal yield of the collagen peptides [100]. The concentration has a direct impact on the pH value that changes the electrostatic interaction and the structural conformation. The pH further dictates the extraction capacity of a specific collagen peptide from an animal tissue [3]. As shown in Table 4, most researchers apply an incubation temperature of <4 °C with 0.5 M concentration of acid (acetic, citric) for an acid-soluble collagen extraction process [3,83,84,85,86,87,88,89,90,91].

### 4.2. Enzymatic Hydrolysis 

Chemical hydrolysis is traditionally used by most manufacturers; however, enzymatic hydrolysis is preferred due to its various benefits over acid hydrolysis. For example, enzymatic hydrolysis is used to upregulate various functional and nutritional properties of the collagen obtained from different species. The proteolytic enzymes (proteases) catalyze the proteolysis and help to break the triple helical peptides into small and individual peptide chains. Various proteolytic enzymes are reported to perform the catalytic proteolysis reactions, for example, alcalase, pepsin, trypsin, chymotrypsin, papain, pancreatin, bromelain, properase E, nutrase, flavourzyme, protamex, etc. [92,93,94,95,96,97], as shown in Figure 1. The selection of a particular enzyme is based upon the targeted peptide residue in the substrate that is defined as the “enzyme’s specificity”. The recent adaptations show the combination of both chemical and enzymatic hydrolysis for the extraction of collagen peptides [3,25,82]. One prominent work reports the use of combined acetic acid for acid hydrolysis followed by enzymatic hydrolysis by pepsin (albacore tuna) [83]. The combined acidic and enzymatic hydrolysis (Figure 4) for long incubation times results in a high yield while keeping the temperature the same as that of acid hydrolysis (4 °C) as noted in Table 4. Recent research work also implements sequential dual enzymes-based hydrolysis, which reports a good yield from sediments of marine sources (gutted fish) [56].

### 4.3. Pre-Treatments 

Collagens are thermo-sensitive biomaterials. The thermal sensitivity of different types of collagens obtained from different sources depends on the conformational restrictions by pyrrolidine rings of hydroxyproline and proline (amino acids). The hydrogen bond shared by the hydroxyl group of hydroxyproline also contributes to the thermal stability of the helical structure [99,102]. The chemical hydrolysis of different raw materials is preceded by alkaline or acidic treatment, or both, each of which reports specific advantages. For example, the alkaline pre-treatment extracts the unwanted products such as non-collagenous proteins and pigments, and reduces the endogenous effects of protases [103,104]. In this regard, two alkali solvents are reported extensively, i.e., NaOH and Ca(OH)_2_ [105]. NaOH is preferred over Ca(OH)_2_ because it causes comparatively more swelling to assist the subsequent chemical extraction process through promoting the mass transfer rate in a tissue matrix [99]. 

Furthermore, the time, concentration and temperature of alkaline and acidic solvents to separate the unwanted non-collagenous components are vital processing parameters in the pre-treatment process. Liu et al. observes the direct proportional relationship between non-collagenous extracts with the increase of time, temperature, and concentration [99]. The optimal values of time, temperature, and concentration of alkaline pre-treatment are considered vital as the degradation of collagen peptides is observed at extreme temperatures for long exposure times of treatment in SDS-PAGE patterns of proteins [99]. Sato et al. investigates the effects of different concentrations of NaOH for the pre-treatment of the collagen source (fish myocommata) [103]. The results show degradation of collagen peptides after 72 h of pre-treatment at 1.0 M and 5 °C of NaOH. Hou et al. presents a 27-3 fractional factorial design of experiment (DoE) to investigate the effects of extreme concentrations and temperature on collagen degradation [106]. 

The combination of alkaline, acidic, and enzymatic pre-treatment are also reported with high collagenous protein extraction in the pre-treatment stage. Recently, Mei et al. combined the alkaline and acidic pre-treatment with enzyme pre-treatment before proper enzymatic hydrolysis [25]. They utilized the alkaline proteases to remove fatty acids (non-collagenous protein) from Salmo salar skin. The significant difference of using all three techniques as pre-treatment is the rapid breakdown of non-collagenous products in less than an hour as compared to combined acidic/alkaline or individual acidic/alkaline pre-treatments [25]. Nalinanon et al. uses NaOH initial pre-treatment and then butyl alcohol for removing fatty acids from collagens during the pre-treatment stages [83]. Based on well-researched numbers for temperature, concentration, and time, most of the recent researchers utilize 0.01–0.1 M of alkaline solvent and 0.05 M of acidic solvent for acidic pre-treatment (cleaning) as shown in Table 5.

### 4.4. Ultrasound-Based Extraction

Ultrasound treatment of collagen raw sources is the most common non-destructive and non-invasive method for extracting collagen peptides or hydrolysate. The ultrasound waves cause thermal, chemical, and mechanical changes to the substrate (proteins/collagens/enzymes) due to the formation and abrupt collapse of cavitation bubbles. The cavitation generates strong shear forces that break the covalent hydrogen bonding in the substrate. The rate of cavitation is controlled by the intensity of the ultrasonic waves. The hierarchy of different research approaches associated with ultrasound treatment includes the following four sub-categories: (a) power, (b) intensity, (c) amplitude, and (d) exposure time (Figure 5). 

Power and intensity are two important parametric aspects that were used with great care in the early utilization of ultrasound for collagen extraction. One of the first research projects on the extraction of collagen hydrolysate reported mild power waves (120 W) with high intensity (40 KHz) to treat the bovine tandem collagen in the presence of a pepsin enzyme. The results report a significant increase of yield to 88% as compared to 71.4% of pepsin enzymatic soluble extraction after two days of treatment [107]. In another research project, non-collagenous proteins (trypsin) are treated with mild ultrasonic waves in the presence of proteolytic enzymes. The rationale for using mild waves is the negative impact of high-power waves (26.4 kHz and 26 W/cm^2^ [108] and 100 to 500 Watt 20 kHz [109]) [109]. Recent developments in ultrasonic treatment report the use of a combination of high power (>100 Watt) and intensity (>20 kHz) for ultrasonic treatment. Amiri et al. evaluates the effects of variable power (100 and 300 Watt) for variable times (10, 20, 30 min) on the Longissimus dorsi muscle from five Holstein bulls [110]. High power treatment at 300 Watt for 30 min produces small particle sizes that provide significantly enhanced pH, emulsification properties, water-holding capacity, and gel strength [110]. The research has introduced the optimal management of power and time for future applications. However, it does not provide any effect on yield, which will be a novel finding for the future [110]. Another recent research project by Akram et al. reports the highest power of 950 Watt and intensity (20 to 25 kHz) with an astonishing long interval of 6 to 36 min to extract collagen type II from chicken sternal cartilage [32]. The research is unique as it raises key future questions on physiological and biological mechanisms behind the resilience of chicken cartilage to extreme ultrasound treatment [32]. 

Amplitude is another factor that is used along with an optimal combination of power and intensity. The high-intensity ultrasonic extraction is performed at 20 kHz at variable amplitudes (20%, 40%, 60%, and 80%) with a pulsed interval of 20 s for 24 h on sea bass skins (Lateolabrax japonicas). The results report a yield of 93% from acetic acid combined with ultrasonic treatment as compared to 20% of a just acid soluble extraction [111]. 

Another aspect of ultrasonic collagen extraction is investigation of the optimal time of ultrasonic treatment to avoid the common denaturation of collagen enzymes/proteins. To extract collagen from any source, it is necessary to break the covalent bonds of lysine and hydroxylysine that form the inter- and intra-molecular cross-links. However, long intervals of exposure to ultrasonic waves (regardless of intensity) cause high shear, temperature, and pressure inside the medium of the raw source [112]. This leads to a denaturation of obtained collagen extracts due to the breakage of the hydrogen bond and Van der Vaals forces in the chains of polypeptides [111]. This produces a certain type of collagen (over-denatured products). Ran et al. introduces the use of ultrasonic waves for a minimum (optimal) time accompanied with pre-acid (acetic acid) and post-enzymatic (pepsin) treatment to extract non-denaturized proteins from cattle tandem [112]. The research uses a novel idea of maintaining the temperature at 4 °C during ultrasonic treatment through a water-cooling bath which helps to overcome the over-denaturization. The results show a high rate of yield (5.7%) for 0 to 12 h of ultrasound treatment as compared to 6.2% for 0 to 24 h [112]. 

This concludes that the set of values associated with power, intensities, and time is based on different sources, which is not yet explored for many contemporary collagen species. This is also summarized in Table 6. Therefore, it highlights another research question to investigate the optimal ultrasonic parameters for unexplored raw sources.

### 4.5. Super-Critical Carbon Dioxide-Based Extraction

All techniques reported earlier have some drawbacks regarding environment. Barros et al. presents a novel green extraction technology for collagen/gelatin extraction [113]. Carbon dioxide pressure (50 bar) provides the acidic reaction in the extraction process. The study has achieved a yield of about 50% for marine sponges [113]. Later on, Silva et al. extended the novel technique through providing a detailed investigation of optimal parameters of time (3, 13.5, 24 h) and pressure (10, 30, 50 bar) [114]. The results provided an improvement in yield with an increase in time of extraction [114]. The increase in pressure of carbon dioxide is still a mystery to be solved, and hence it highlights a novelty to explore in the future. Furthermore, this extraction process is explored for the least of all other processes shown in Figure 1. Therefore, based on green technology prospects, it reveals a potential research field to explore in collagen extraction processes.

### 4.6. Hydrostatic Extraction

Another key extraction method is hydrostatic extraction at variable temperatures, pressures, or both. The hydrostatic extraction is also preceded or followed by: alkaline, acidic, or enzymatic treatment as shown in Table 7 and Figure 1. 

#### 4.6.1. High Pressure and Temperature with Acid or Alkaline Pre-Treatment

In the early literature, the high-pressure treatment was used as a post-process for inducing gelation properties in the collagen hydrolysate or gelatin [115,116]. Walkenstrom et al. uses the high pressure of 6 kBar for 20 min to improve the rheological and microstructural properties of mixed whey protein and bovine gelatin [115]. The high-pressure gels showed a high degree of aggregation at a pH of 7.5 as compared to heat-treated gels [115].

Moving ahead, the researchers add an additional parameter of temperature along with high pressure to extract collagen. For example, Montero et al. investigates the effects of different pressures and temperatures on gelation properties (turbidity, gel strength, viscosity) of cod and megrim gelatin extracted from a preliminary process of acid solubilization (Table 7) [116]. The cod shows high gel strength as a result of high pressure as compared to simple acid-treated gelatin [116]. 

Later on, the literature regarding hydrostatic extraction includes the acid pre-treatment followed by high pressure and temperature extraction. For example, Gómez-Guillén et al. registers the first attempt to extract collagen gelatin with high pressure (250 MPa, 400 MPa) and temperature for variable times (10, 20 min) assisted with mild acetic acid pre-treatment [117]. Although the extraction has shown a slight improvement (22.8%) as compared to pure acid soluble collagen (21.3%), the molecular weight distribution in SDS-PAGE patterns and the time for extraction are significantly improved for high pressure extracted collagen [117]. 

The literature also presents the pre-alkaline and acidic treatment followed by high temperature water extraction of giant squid collagen [118]. Although the yield (7.5%) was not significant as compared to high pressure/temperature applications reported earlier [117], the research provides valuable information regarding improvement in properties with combined alkaline, acidic, and water treatment at only high temperatures [118]. One of the reasons for such a low yield is insufficient time for water treatment [119], which is a future research opportunity to explore as it has not yet been performed for giant squid.

#### 4.6.2. Water Extraction

The hydrostatic extraction is also performed with the aid of “water”. In this regard, Kołodziejska et al. reports one of the first research works on the effects of variable time (15 to 120 min) and temperature (45, 70, 100) in pure water for skin and back bones of fish and cod, respectively [119]. The authors found an astonishing 100% yield of gelatin at 45 °C after 15 to 60 min of the extraction procedure [119]. 

One of the modifications of the water extraction technique is recently reported with the addition of post-enzymatic hydrolysis. In this regard, the most recent research was performed on pig skin using high temperature water (100 °C) to extract the gelatin for scopolamine which causes the impairment of cognitive functions [21]. The authors compare the two kinds of extraction procedures: (1) water extraction at high temperature, and (2) combined water extraction at high temperature followed by enzymatic hydrolysis. The combined extraction at high dosage (100 and 400 g/day) shows better results in the form of increased latency time, increased acetylcholine (ACh) content, and decreased ACh esterase activity. This is also one of the few papers that reports water/enzyme treatment [21].

The discussion in this section concludes the novelty to explore the effects of (1) water extraction accompanied by enzymatic hydrolysis for different species, and (2) water extraction accompanied by combined pre-chemical and post-enzymatic hydrolysis for different species.

## 5. Disease

The research on collagen is also aligned to the cure or attenuation of symptoms associated with various animals or human impairments/diseases (Figure 6). Based on the research related to collagen Types I and II in abundance, the main types of medical impairments include: osteoarthritis (OA) pain, osteoporosis, bone regeneration, bone dysplasia, cognitive functions, hypertension, anti-aging, skin-related anti-photoaging, skin-related anti-oxidation, wound healing, and metabolic syndrome. This section includes the details of medical ailments selected by various researchers to report for collagen/collagen peptides/hydrolysate-based cures.

Kimura et al. reports the cure of dysphonia due to paralysis in the vocal folds with collagen [48]. The collagen as a supplement is injected into the arytenoid, resulting in improved voice function and quality for patients with unsatisfactory glottal competence [48].

Lee et al. cures a pressure ulcer with collagen protein hydrolysate [121]. The eight weeks of treatment with collagen supplement along with proper clinical care of 71 residents receives a significantly high PUSH (pressure ulcer scale for healing) score [121]. 

Hays et al. investigates the balance of nitrogen excretion and body weight in elderly women suffering sarcopenia [49]. Sarcopenia is a medical ailment related to muscle loss with age due to the increase in body fats and decrease in bone mass and basal metabolic rate [122,123]. The research provides low excretion of collagen dietary intake with decreased fats in elderly women patients after 15 days of oral ingestion [49].

Diabetes mellitus (DM) is one of the complex metabolic syndromes that is also treated with novel bio-peptides of collagen extracted from animal and food sources [1]. There has been good research on antidiabetic collagen hydrolysate in the past [124,125,126,127,128,129]. However, a fewer number of publications in 2020 to date have reported on this. A recent review on collagen peptides for antidiabetic patients presents a unique area to understand the progression and mechanism of collagen-treated diabetes [1]. Recently, Jin et al. identified a novel peptide for inhibition of an antigenic enzyme (dipeptidyl peptidase IV, DPP-IV) in salmon salar fish collagen for diabetes [130]. The six hydrogen and eight hydrophobic interactions in molecular docking result in the bio-chemical interaction of LDKVFR and DPP-IV. This bio-chemical interaction shows the high inhibitory activity of DPP-IV [131]. Gong et al. isolates the antidiabetic peptides from sea cucumbers (Stichopus japonicus) in gastrointestinal ingestion [131]. The novel 58 peptides reveal high DPP-IV inhibition potency in both gastric (IC50 0.51 mg/mL) and intestinal (IC50 0.52 mg/mL) digestion [131]. Furthermore, the peptides improve glucose and insulin resistance [131].

Osteoporosis is the most common medical impairment investigated in collagen-based research. Recently, Landolo et al. proposed a unique therapeutic approach for osteoporosis [132]. A unique combination of smart conductive materials (PEDOT:PSS) with collagen Type I promotes electromechanical impedance to cause differentiation of neural crest-derived stem cells for novel biometric scaffolds [132]. El-Sayyad et al. highlights bone regeneration (tissue engineering) through achieving improved osteoid tissue for combined Solcoseryl and DM Bone collagen [9]. Zhu et al. reports that porcine bone collagen accelerates the osteoblastic bone formation through cell proliferation of cyclin-dependent protein kinases (CDK2, CDK4), cyclin B1, and cyclin D1 [19]. The research shows significant osteoblast proliferation to overcome osteoporosis [19]. Wu et al. introduces the phosphorylation of a collagen peptide to increase the calcium binding capability of collagen for bones suffering osteoporosis [20]. The collagen peptides show improvement in three of four stages of MC3T3-E1 osteoblasts (proliferation, differentiation, and mineralization) through upregulation of osteopontin, alkaline phosphate (ALP), osteocalcin, and a run-related transcription factor 2 (Runx2) [20]. Another recent effort reports a series of projects on the antagonistic effects on osteoporosis [12,13] for a novel collagen-treated serum with bone turn over biomarkers [12]. The research highlights the reason to use bone turn over bio markers for detection of the pathogenesis of metabolic-caused bone osteoporosis as well as to analyze the therapeutics of osteogenic collagen or drugs [133,134,135,136]. Almost half of detected metabolites (20) in UPLC/Q-TOF-MS analysis shows the regulation of primary amino acid-based and lipid-based metabolism that recover the serum to its normal values for osteogenesis [12]. The research on osteoporosis is also aimed with respect to a particular subject, whose medical condition reports the same disease with different levels of severity [137]. A similar kind of research is reported for bone loss in post-menopausal women. The research shows significant inhibition in vitro bone formation for calcium-chelated collagen hydrolysate [137].

Rheumatoid arthritis pain [138] is another orthopedic-based topic that is reported to have been cured with collagen peptides [8]. Lu et al. resolve the knee OA pain in a comparative study with glucosamine hydrochloride mixed with chondroitin sulfate [68]. The collagen Type II shows a high Western Ontario McMaster (WOMAC) Universities’ Osteoarthritis Index with less pain [68].

Depression and hypertension are important medical impairments that are extensively investigated with collagen hydrolysate [41,139,140,141,142,143,144,145,146,147,148]. Recently, a couple of efforts by the same group were presented on diminution of hypertension using novel collagen hydrolysate [41,149]. The group proposes that the presence of prolyl-hydroxyproline (PO) [41,149] and hydroxyprolyl-glycine (OG) [41] decreases the expressions of depression in a forced swim test [41]. Another recent research proposes a novel hydrolysate source (mung bean) for mitigation of hypertension [36].

Collagen hydrolysates are also investigated for cognitive impairments such as Alzheimer’s disease (AD) or dementia. The pathogenesis of this neurodegenerative disease includes dysfunctional neurotransmitters, amyloid beta peptides, etc. These pathogeneses cause a high expression of the following three syndromes: (1) oxidative stress, (2) inflammation, and (3) nervous dysfunction. Most of the collagen-based literature targets the antioxidative stress through different therapeutic ways [150,151]. Oxidative stresses cause cell death due to the mitochondrial dysfunction, apoptosis, and neuronal damage. One of the scientific ways to attenuate oxidative stress is the activation of corresponding neuronal stem mesenchymal cells through signaling pathways. As a post-neuronal response to cell death, different signaling pathways activate to proliferate the new cells against oxidation, inflammation, and neuronal dysfunction [150,152]. These pathways include mitogen-activated protein kinase (MAPK) and phosphoinositide 3 kinase (PI3k)/protein kinase B (Akt) pathways. Another important pathway is the nuclear-related factor 2 (nrf2) defense pathway [150,153]. As a response to oxidative stresses, reactive oxygen species (ROS) release the nrf2, which causes a nuclear translocation interaction with the antioxidant response element (ARE). ARE interactions cause a cytoprotective effect against the oxidative activity along with the protection of anti-oxidative enzymes/proteins present in the body [150]. Most of the collagen-based research aims for the cure or attenuation of dementia or Alzheimer’s disease through the abovementioned activation, proliferation, and differentiation of the above-mentioned mesenchymal stem cell through corresponding neuronal signaling pathways.

Zhang et al. reports one of the few studies recently on an improvement with respect to AD through nrf2 defense signaling pathways, the Bax/Bcl-2-related anti-apoptosis pathway, and the CREB-related neuronal survival pathway using round scads hydrolysate (RSH) [150]. The research reports the mechanism of the binding of keap1 (Kelch-like ECH-associated protein 1), a repressor protein to sense the disturbance in cellular homeostasis, to control the nrf2 oxidative defense pathway against glutamate-induced oxidation in the PC12 cell line. PC12 cell lines are cultured in a laboratory environment that contains pheochromocytoma (tumors) from the adrenal medulla of rats. This helps to alleviate the mitochondrial dysfunction, inflammation and neuronal oxidation [150]. Koizumi et al. reveals the alteration in brain structure against dementia with the collagen hydrolysate intake [22]. Chan et al. concludes with a significant decrease in memory loss, shrinkages in the dentate gyrus area, and neurodegeneration with the patent collagen hydrolysate [151]. The collagen hydrolysate with antioxidant amino peptides and anserine also results in the downregulation of amyloid β peptides [151]. Kim et al. utilizes the novel collagen hydrolysate to recover the cognitive impairment clinically induced with scopolamine injections in mice [21]. The collagen hydrolysate increases the levels of cholinergic neurons through a mechanism that increases the neurotransmitter of acetylcholine (ACh) and decreases the acetylcholine esterase (AChE) level [21]. The proposed mechanism results in the control of dementia or AD [21].

## 6. Receptors-Based Collagen

Collagen hydrolysate has been reported with a number of non-toxic amino acids that have various advantages, e.g., non-toxic supplements, biomedically compatible and viable properties, nutraceutical implementations, therapeutics for skin-recovery, wound healing, memory gain, bone formation, brain structure, etc. These positive syndromes are achieved through a corresponding biomedical mechanism at the molecular and cellular level. Briefly, it includes proliferation, differentiation, and formation of any mesenchymal stem cell (osteoblast, fibroblast, chondrocyte, adipocytes), which are based on the activation of appropriate biomedical receptor or signaling pathways (metabolic system) as shown in Figure 7 and Table 8. The work on collagen hydrolysate regarding these cellular signaling pathways is registered either as the sole target or incorporated as a part-analysis to detect the main reason of any improvement (osteogenesis, anti-aging, anti-photoaging, wound healing, cognitive functional improvement).

The P75 neurotrophin receptor (p75NTR) or p75 nerve growth factor receptor (NGFR) or CD271 are mesenchymal stem cell markers [154,155,156] that are expressed in non-neuronal cell types (fibroblasts, macrophages [157], chondroblasts, adipocytes, osteoblasts [158,159]) during inflammation in the skin caused by any wound [157]. Asai et al. [160] reports the biomedical functionality of Prolyl-hydroxyproline (Pro-Hyp) collagen through activation of P75 receptors. The authors find the p75NTR cells using immunocytochemical staining in migrated fibroblasts from mouse skin even after a prolonged incubation time [160]. The expression levels of p75NTR fibroblasts raised significantly on Prolyl-hydroxyproline (Pro-Hyp) collagen incubated samples showing the cell differentiation for wound healing [160]. The P75 neurotrophin receptor (p75NTR) signaling pathway is provided in Figure 7a.

PI3K/AKT is an intracellular signal pathway to regulate cell growth, proliferation, differentiation, angiogenesis, and apoptosis through any one, or a combination, of its main composing constituents: (a) phosphatidylinositol-3,4,5-bisphosphate (PIP3), (b) tyrosine kinase (RTKs), (c) phosphatidylinositol-4,5-bisphosphate (PIP2), (d) phosphatidylinositol 3-kinase (PI3K), and (e) AkT/protein kinase B [19,161,162,163]. The PI3K/AKT pathway is shown in Figure 7b. Zhu et al. [19] biologically activates the PI3K/AKT pathway by low molecular weight tripeptides of porcine collagen that cause downregulation of PTEN and upregulation of p-Akt, a cyclin dependent protein kinase (CDK-2, CDK-4). This leads to cell proliferation, differentiation, and anti-apoptosis of osteoblasts [19].

Pattern recognition receptors, PRRs (such as NOD2)m and factor BD14 are involved in nucleotide oligomerized binding at the wound site that cause phenotype regulation of inflammation and wound microorganism colonization. Under a wound-healing environment, NOD2 activates the NF-κB signal pathway through a special kind of receptor (Toll-like receptors, TLRs), that upregulate different anti-bacterial factors (BD14) [25,164,165,166,167]. Mei et al. [25] biomedically cures the wound through activation of PRRs (NOD2) and BD14 using marine extracted collagen. The decrease in levels of expression of proinflammatory cytokines (TNF-α, IL-6, and IL-8) and the increase in expressions of an anti-inflammatory cytokine (IL-10), the vascular endothelial growth factor (VEGF), fibroblast growth factors (β-FGF), and NOD 2 present significant wound healing [25]. The signaling pathway for NOD2 receptors is depicted in Figure 7c.

Transforming growth factors (TGF-β) are multifunctional cytokines in the super family of transforming growth that are regulated by a Smad signal pathway. One of the main intracellular functions by TGF-β is homeostasis to attenuate skin aging. During non-regulated cell death because of skin aging, the TGF-β combines with TGF-β Type II receptors (TβRII). The serine/threonine kinase activity of the TGF-β Type I receptor stimulates the phosphorylation and activates the Smad signal pathway (Smad 2 and Smad 3). This achieves the biosynthesis of new cells [168,169,170,171,172]. Liang et al. [172] reports the reduction in skin aging with collagen hydrolysate through upregulation of Type I mRNA, III mRNA, and TβRII through activation of Smad signaling pathways. The research also reports decreases in biological expression levels of skin oxidation and matrix Metalloproteinase (MMPs) [172]. The transforming growth factor (TGF-β) in the SMAD signaling pathway is shown in Figure 7d.
materials-14-02806-t008_Table 8Table 8Literature with respect to receptors and pathways.Collagen SourceReceptor/Proteins/ComplexActivated PathwayAssociated Receptors with Activation of PathwayResulting Factors(Anti-Bacterial, Antimicrobial, Anti-Oxidation)Biomedical EffectProlyl-hydroxyproline (Pro-Hyp) [160]p75NTR---Wound healingPorcine collagen [19]PTEN,CDK-2,CDK-4PI3K/AKT

Cell proliferation,anti-apoptosisSalmo salar skin collagen peptides (Ss-SCPs) and Tilapia nilotica skin collagen peptides [25]pattern recognition receptor (NOD2)NF-κB signal pathwayToll-like receptors (TLRs)BD14 (modulation through upregulation/down regulation/null)Wound healingChum Salmon skin hydrolysate [172]TGF-βSmadTGF-β type II receptors (TβRII)Type I and III mRNA,TGF-βRIISkin cellular homeostasisPorcine bone collagen peptide [20]Runx2MAPK--Osteoblastβ-CateninWnt/β-catenin--Fish collagen hydrolysate [173]IL-1βERK and p38-MMP13Osteoarthritis

Run-related transcription factor 2 (RUNX2) receptors with a Runt DNA binding domain is capable of cell proliferation in the G1 stage of the cell cycle. These sp6 human gene factors transcript the initial commitment for osteoblast proliferation followed by sp7 and Wnt-Signaling [174,175]. The RUNX2 is activated by MAPK (mitogen-activated protein kinases) or the ERK (extracellular signal-regulated kinases) pathway to transcript the osteogenesis [176,177]. Wu et al. [20] maps the increase in osteoblast expression in the MC3T3-E1 cell assay due to phosphorylated collagen hydrolysate with upregulation of RUNX2 and β-Catenin proteins. The upregulation of RUNX2 and β-Catenin is inferred based on the literature on the activation of MAPK and Wnt-Signaling pathways [20,178,179].

Mitogen-activated protein kinases (MAPK) are one of the main serine/threonine kinases that create a signaling-based stimulus against ultrasensitive responses for various cellular functions such as cell proliferation, differentiation, apoptosis, inflammation, metabolism, memory, etc. There are three main signal pathways for the beforementioned effectors to initiate: (1) the extracellular-signal-regulated kinase (ERK) pathway, (2) c-the Jun N-terminal kinase (JNK) pathway, and (3) the p38 pathway. The p38 pathway is activated by interleukin cytokines (IL-1β) to transcript a signal for inflammation in cartilage osteoarthritis. Boonmaleerat et al. [173] present the activation of p38 and p-ERK through IL-1β to cause inflammation in the cartilage synovial joint. This is one of the rare research projects that highlights the acute effects of the different size of fish collagen hydrolysate to raise the levels of matrix metalloproteinase (MMP13) in cartilage metabolism [173].

As a short summary, the collagen intake (oral or non-oral) stimulates different receptors or pathways. The most common receptors recently reported are the p75 nerve growth factor receptor and pattern recognition receptors, etc. Similarly, few of the recently reported pathways include the Smad signal pathway, PI3K/AKT intracellular signal pathway, etc. These stimulations of corresponding receptors, factors, or pathways helps to achieve different biomedical and biological functionalities. In this regard, commonly reported functionalities include wound healing, cell proliferation, anti-apoptosis, bone reformation (osteoblasts proliferation), homeostasis, etc. 

## 7. Future Perspective and Challenges

Based on the details noted in the above-mentioned five categories, a few of the major areas are found potentially viable for future research. These are mentioned in Table 9. 

It is noted that most of the novelties highlighted in the Table 9 are related with the lack of experimenting on different species with different processing techniques. A basic rule of thumb in biology is important to consider here, as each species has a different structure (bone and muscle) and bone mineral density [180]. Furthermore, the composition of cortical and cancellous bones is different among species [180], which can be a major factor for extracting high molecular biopeptides.

Due to the difference in composition and structure [180], the species compatibility to a particular process is a big challenge. The ultrasonic treatment at high parameters (intensity, power, and amplitude) for long processing intervals can degrade the true biopeptide sequences (collagen). 

Similarly, the efficiency of the water extraction along with pre-chemical and post-enzymatic hydrolysis depends on the endurance of a specific type of species. Each species is comprised of a different biological anatomy that can endure a particular level of extraction parameters. The biological factors can play a significant role in determining the optimum quantity (yield) and true quality (biopeptide sequences) from the extraction process regardless of severe pre-chemical and post-enzymatic treatments.

The sediment extraction method provides better yield as compared to the acid solubilized collagen (ASC) in the case of fish sediments. In our point of view, one of the reasons for better yield with high molecular weight biopeptides sequences is the compatibility of fish with the process. However, the processing of sediments from other species such as bovine, porcine, etc. may provide different challenges to achieve better yield than ASC. In this regard, the combination of the sediment method with acid or alkaline pre-treatment may catalyze the sediment process to achieve good yield. 

## 8. Conclusions

Collagen is one of the most versatile biopeptides with a high protein value. The collagen extracted from various species are hydrolyzed to collagen hydrolysates. The review highlights various benefits of collagen intake (oral ingestion, inoculation, surgical implant). The benefits are associated with the development or enhancement of biological, biochemical, and biomedical functionalities. The review presents a novel hierarchy that enlists different functionalities within five key aspects (processes, Type, species, disease, and receptors) that are not yet systematically covered in the literature. Particularly, the receptors are rarely reviewed for the effects of collagen intake. The novel hierarchy purposefully provides a simple but detailed outlook of the decades of research on collagen. The review contributes by highlighting various research novelties based on specific aspects. Furthermore, the review points out the challenges associated with the biological composition and structure of different species.

## Figures and Tables

**Figure 1 materials-14-02806-f001:**
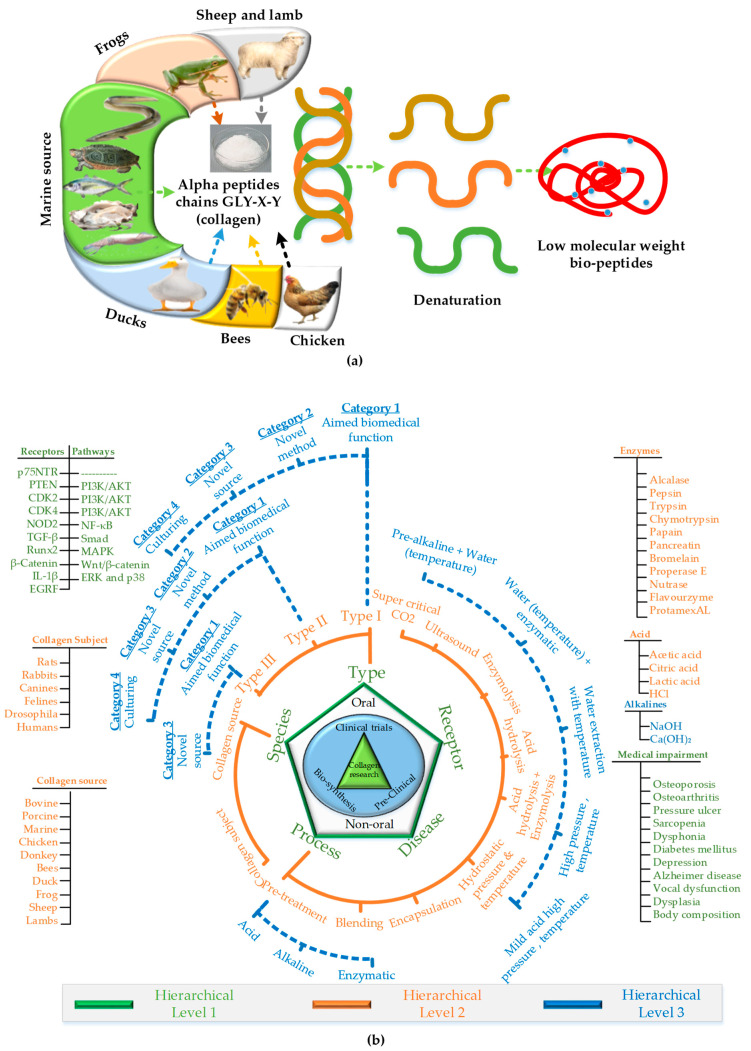
Collagen and collagen hydrolysates: (**a**) Schematic of collagen processing, and (**b**) Novel research hierarchy on collagen or collagen hydrolysate or gelatin based on observed literature.

**Figure 2 materials-14-02806-f002:**
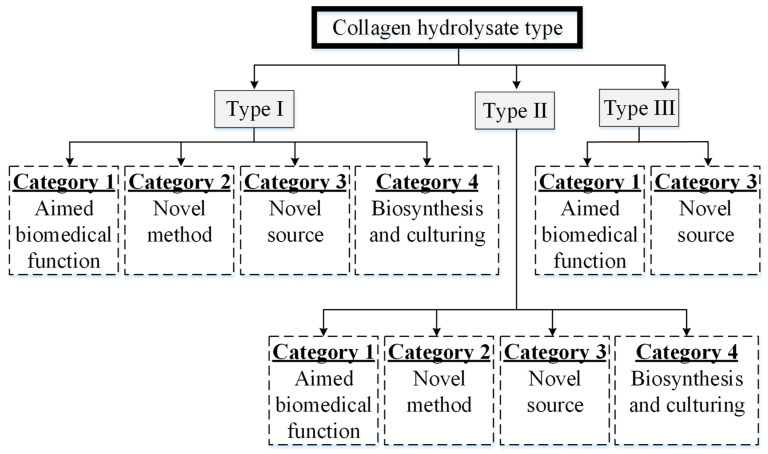
Hierarchical segregation with respect to the “Type” of collagen or collagen hydrolysate or gelatin.

**Figure 3 materials-14-02806-f003:**
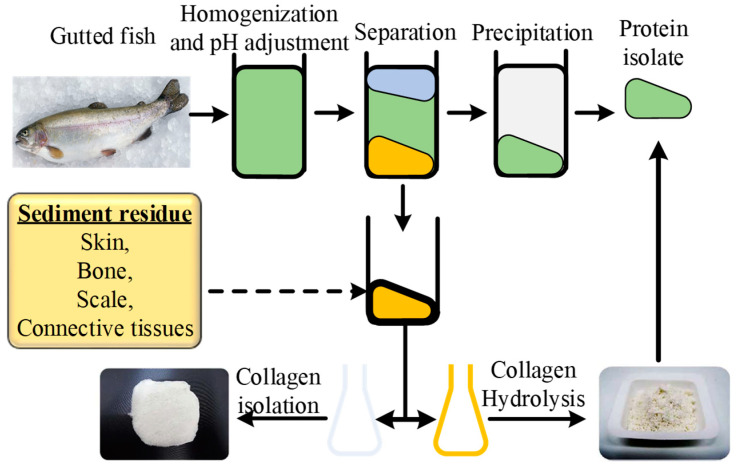
Sediment method for collagen extraction.

**Figure 4 materials-14-02806-f004:**
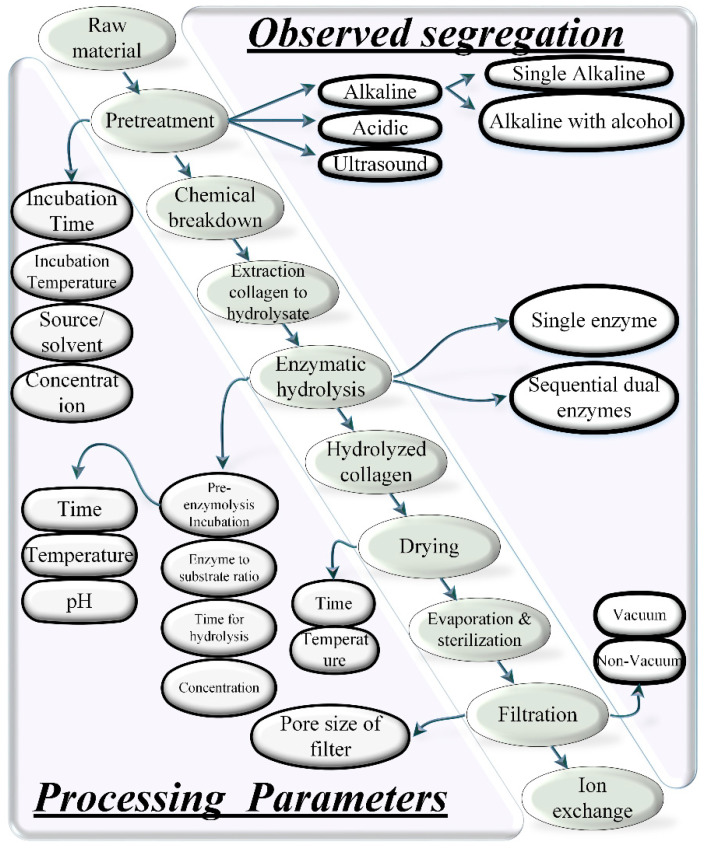
Parameters and categories of the combined chemical and enzymatic hydrolysis.

**Figure 5 materials-14-02806-f005:**
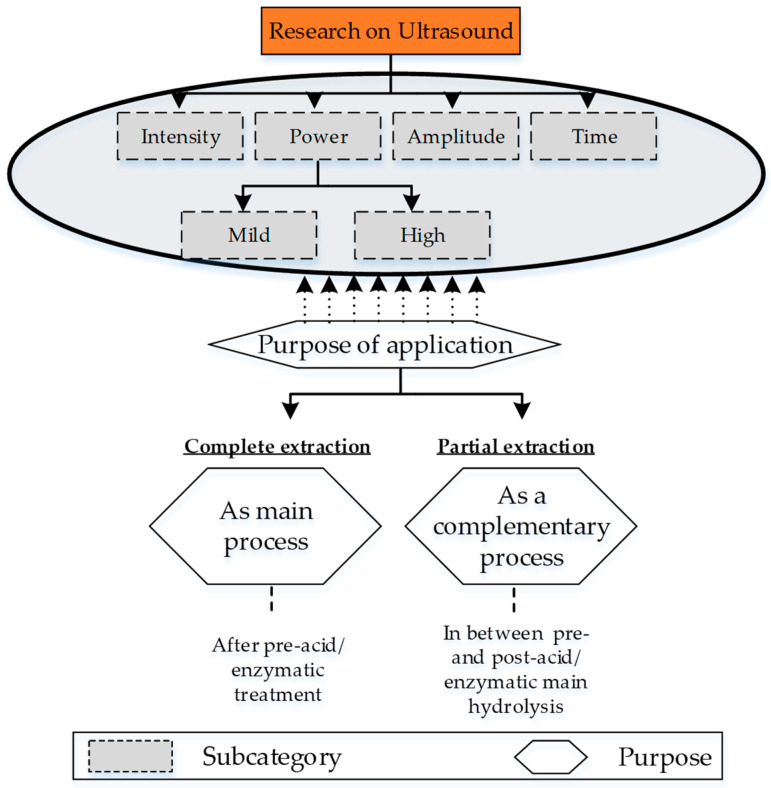
Hierarchical segregation for ultrasound-based collagen extraction.

**Figure 6 materials-14-02806-f006:**
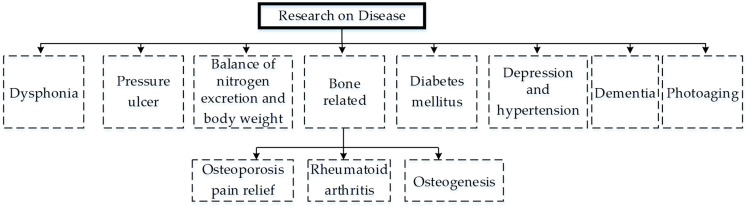
Hierarchical breakdown for research on disease (the dark rectangle shows the level 1, and the dotted rectangles are showing subcategory).

**Figure 7 materials-14-02806-f007:**
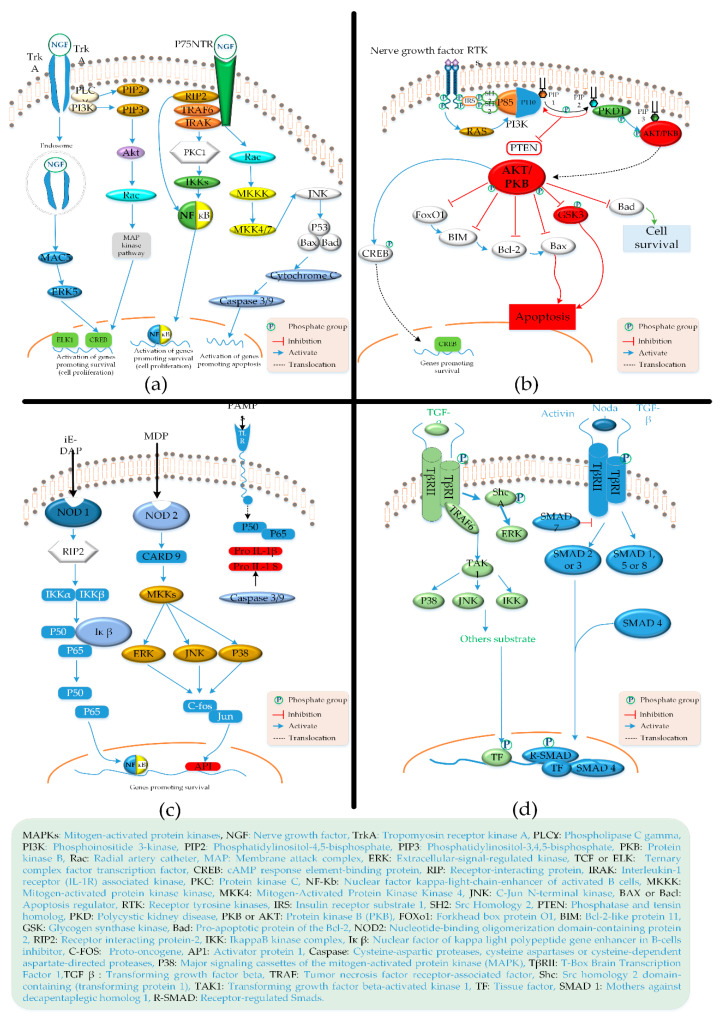
Signaling pathways and corresponding receptors: (**a**) P75 neurotrophin receptor (p75NTR) activation pathway, (**b**) PI3K/AKT is an intracellular signal pathway, (**c**) pattern recognition receptors (NOD1, NOD2), and (**d**) transforming growth factors (TGF-β) in SMAD signaling pathway.

**Table 4 materials-14-02806-t004:** Acid- and enzyme-based hydrolysis with common process parameters.

Collagen Source	Acid Solvent	Incubation Time (Hours)	Concentration (M)	Incubation Temperature (°C)	Ratio of Source to Solvent	Yield (%)
Octopus, outer skin [85]	Acetic acid	2 times 24	0.5	4		5
	Acetic acid with pepsin	48	0.5	4		50
Cod, skin (minced) [84]	Acetic acid	24	0.1	4	1:6	52
	Acetic acid	24	0.25	4	1:6	54
	Acetic acid	24	0.5	4	1:6	59
Cod, skin (whole) [84]	Acetic acid	24	0.5	4	1:6	20
	Citric acid	24	0.5	4	1:4 to 1:20	10 to 25
	Citric acid	3 times 24	0.5	4	1:4 to 1:20	70 to 90
Freshwater Carp, bones [91]	Acetic acid	24	0.5	4		20
Freshwater Carp, scales [91]	Acetic acid	24	0.5	4		7
Japanese sea bass, bone [87]	Acetic acid	72 + 48	0.5	4		41
Japanese sea bass, skin [87]	Acetic acid	72 + 48	0.5	4		51
Freshwater Nile percha (adult), skin [101]	Acetic acid	16	0.5	15		59
Freshwater Nile percha (young), skin [101]	Acetic acid	16	0.5	15		63
Ocellate puffer fish, skin [86]	Acetic acid	72	0.5	4		11
	Acetic acid with pepsin	48	0.5	4		45
Sardine, scales [90]	Acetic acid	-	0.5	4		5
	Acetic acid with pepsin	96	0.5	4		14
	Acetic acid with pepsin	96	0.5	15		71
Shark, cartilage [88]	Acetic acid	-	0.5	4		20
	Acetic acid with pepsin	-	0.5	4		54

**Table 5 materials-14-02806-t005:** Pre-treatment processes (alkaline, acidic, enzymatic) with common parameters. “α” means directly proportional.

Collagen Source	Solvent	Pre-Treatment			Optimal Pre-Treatment Values			Relation of Yield
		Time t (Hours)	Concentration C (Moles)	Temperature T (°C)	Time (Hours)	Concentration (Moles)	Temperature (°C)	
Grass carp skin [99]	NaOH	1, 2, 4, 8 or 12	0.05, 0.1, 0.2 or 0.5	4, 10, 15 and 20	12 h	0.05 and 0.1 M Increased loss of Hydroxyproline Above 0.1 M	4 and 20	α t, α C, α T
	Acetic acid	72	0.1, 0.2, 0.5 or 1.0	4, 10, 15 or 20	72	0.5	Similar in range 4–20 °C at 0.5 M	α C (<0.5 M), α T (<0.5 M)
Fish myocommata [103]	NaOH	96 (4 nights)	0.01, 0.05, 0.1, 0.5, 1	5	96	0.1	5	α C
Alaska pollock cold water fish skin [106]	Ca(OH)_2_	0.75 (45 min) and 1.5	0.01 and 0.1 mol/L of OH in alkali solvent	2 and 20	Not clear	0.1	2 °C was noted for significant loss of collagen	α C
Alaska Pollock Skin [105]	NaOH Ca(OH)_2_	1	0.01, 0.1, 0.2, and 0.5 mol/L of OH in alkali solvent	2 to 4	1	0.01 and 0.1	2 to 4	α C (<0.1)
	acetic acid	1	0.05 mol/L	2 to 4	1	0.05	2 to 4	Not investigated
Arabesque greenling [83]	NaOH, butyl alcohol	6 (NaOH) 18 (butyl alcohol) 6	0.1 M/lit	4	Same	Same	Same	Not investigated
Salmo salar skin [25]	NaOH Acetic acid Enzyme	0.16 (10 min)	0.1 M NaOH 0.005 M Acetic acid	Not given	Not investigated	Not investigated	Not investigated	Not investigated

**Table 6 materials-14-02806-t006:** Ultrasonic processing for collagen extraction.

Source	Pre-/Post Ultrasonic Processing/s					Ultrasonic Processing					Yield (%)	Optimal Parameters/Results
	Solvents	Concentration	Temperature (°C)	Time	Ratio of Source to Acid Solution	Power and Amplitude	Intensity	Time	Temperature	Pulse Interval		
Longissimus dorsi muscle from Holstein bulls [110]	Pre-processing KCl, K_2_HPO_4_, KH_2_PO_4_, EDTA	100 mM (KCl), 25 mM (K_2_HPO_4_) 25 mm (KH_2_PO_4_) 2 mM (EDTA)	4		Not given	100, 300	20 kHz	10, 20, 30 (Minutes)	Not given	On-time 2 s Off time 4 s	Not given	300 Wat and 30 min high pH, Good gel strength, Good emulsification
Cattle tandem [112]	Pre-process Acetic acid Post- process Pepsin with 0.5 M acetic acid	0.5 M/liter (Acetic acid) 20–70 U mg^−1^ of tendon (Pepsin + acetic acid)	4	0 to 24 h	1:15 (*w*/*v*)	Low power (amplitude Not given)	20 kHz	0 to 24 h	4	On-time 20 s Off time 20 s	Based on dry weight 6.2% (24 h of US & 24 h of enzyme acidic)	5.7% is the Optimal for 12 h of treatment
Sea bass skins [111]	Pre-process Acetic acid	0.5 M/L	4		1:2 (*w*/*v*)	Amplitude 20, 40, 60, and 80%	20 kHz	0 to 24 h	4	On-time 20 s Off time 20 s	93%	Combined acidic and ultrasonic has better result
Chicken sternal cartilage [32]	Pre-process NaCl, Tris HCl, Na_2_CO_3_, EDTA, Isopropyl Alcohol, Acetic acid	0.05 mol/L (Tris HCl) 0.5 M/L (Na_2_CO_3_) 0.3 M/L (EDTA) 0.5 M/L (Acetic acid)	4	Overnight (NaCl) 24 h (Tris HCl) 24 h (EDTA)	1:10 *w*/*v* (NaCl) 10% *v*/*v* (isopropyl alcohol) 1:6 *w*/*v* (Acetic acid)	950 Amplitude of φ10	20–25 kHz	6, 12, 24, 36 (Minutes)	4	On-time 2 s Off time 3 s	Protein contents excluding other ~84%	Higher the treatment time, higher the yield 36 min is the highest

**Table 7 materials-14-02806-t007:** Water extraction process for collagen extraction.

Source	Category of Hydrostatic Extraction						Hydrostatic Extraction Parameters			
	Pre-Alkaline and Acidic Followed by Water with Temperature	Water + Temperature Only	Water (Temperature) + Enzyme	Acid Base Gelatin then Hydrostatic	Pre-Acid Treatment Followed by Water Extraction	Gelatin with Only High Pressure	Pressure	Time	Temperature	Yield
Mixed Bovine gelatine and Whey protein (Post) [115]						√	6 kbar	20 min		--
Fish skin from Cod, fish skin from megrim (post) [116]				√			200, 300, 400 (MPa)		20 °C, 78 °C	--
Dover sole (Solea vulgaris) skin [117]				√ 0.05 M acetic acid (3 h)				16–18 h	45 °C	--
					√ Mild 0.05 M acetic acid (3 h)		250 MPa	10, 20 min	45 °C	22.8%
Giant squid (Dosidicus gigas) skin [118]	√Skin to water ratio is 1:4							12 h	65 °C	7.50%
Skins of fresh and cold-smoked salmon, Skins of salted and marinated herrings [119]		√						15–120 min	minced skins 45 °C, 70 °C, 100 °C	--
Backbones and heads of cod [119]		√						1st stage 45 min 2nd stage 45 min 3rd stage 45 min	1st stage 45 °C 2nd stage 60 °C 3rd stage 70 °C	--
Channel fish skin [120]	√ Calcium hydroxide solution for 68–76							5.34–6	40 to 46 °C	--
Pig Skin [21]		√						1st stage 7 h 2nd stage 2 h 3rd stage 3 h	100 °C	--
			√					same as before then enzymatic extraction for 12 h	50 °C	--

**Table 9 materials-14-02806-t009:** Novelties concluded in this review.

Category	Novel Areas to Explore for Future
Process	The ultrasound-based extraction at high parameters (intensity, power, amplitude, and time) for various unexplored sources (bovine, porcine).
	Water extraction along with enzymatic hydrolysis for different species is not yet explored.
	Water extraction with combined pre-chemical and post-enzymatic hydrolysis for different species is not yet explored.
	Supercritical carbon dioxide is the least researched technique among all extraction processes in terms of combinations with pre- and post-treatments, species, process parameters (pressure, temperature).
Type	Sediment method is only used to extract Type I collagen hydrolysate from a marine source (gutted fish). It is not yet explored for other types of collagen (Type II and III) and various other species (bovine, chicken, porcine).
	Glucosamine negative effects along with collagen hydrolysate are confusing and need dedicated research for clarifications.
Species	The effects of collagen hydrolysate are not thoroughly investigated for rabbits, cats, and dogs.
	Collagen hydrolysate extracted from Amur sturgeon (swim bladders) does not accumulate in the kidney and may lead to controlled urinary secretions in cats. This needs a separate area of research to discover the potential of novel sources of collagen hydrolysate in cats.
	Different chicken organs (cartilage, heads, skin) as a collagen hydrolysate species are not yet investigated for wound healing, cognitive repair, and antioxidation.

## Data Availability

Not applicable for this review article.

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
