# Peer review of "Developments for Collagen Hydrolysate in Biological, Biochemical, and Biomedical Domains: A Comprehensive Review"

_materials, 2021, doi:10.3390/ma14112806_

Round 1
Reviewer 1 Report
The manuscript presented by M. Harris is an interesting one offering an overview on collagen extraction and applications. The merit of the article is to put together some of the latest information concerning collagen hydrolysate. I also appreciate the structure/classification proposed by the authors.
However the paper, before publication has to be checked/verified from English point of view and errors. For instance I do not think that the expression "research gap" is properly used.
Then there are some notations I do not understand: Why the authors need to write 'Level" (2.6.1, 3.1.2, etc) in my opinion are subchapters.
In table 7 and chapter 4 I do not understand why chitin from squid pens and fish gelatin is mixed with collagen. On the other hand why bovine derma is not included ? As far as I know it is un important source of collagen.
Author Response
Reviewer 1 comments
We are grateful to the reviewer for dedicated efforts that undoubtedly help us to improve our work. We have addressed all comments suggested by the Reviewer. Kindly find the details below,
The manuscript presented by M. Harris is an interesting one offering an overview on collagen extraction and applications. The merit of the article is to put together some of the latest information concerning collagen hydrolysate. I also appreciate the structure/classification proposed by the authors.
However the paper, before publication has to be checked/verified from English point of view and errors. For instance I do not think that the expression "research gap" is properly used.
Authors response
All sentences with the word “research gap” are now rephrased in appropriate manner to highlight research opportunity.
Line 255 to 257: The increase in pressure of carbon dioxide is still a mystery to be solved and hence it highlights a novelty to explore in the future.
Line 297-98: which is a future research opportunity to explore as it has not yet been performed for giant squid.
Then there are some notations I do not understand: Why the authors need to write 'Level" (2.6.1, 3.1.2, etc) in my opinion are subchapters.
Authors response
We have taken corrective actions and removed the word “level” from all subchapters.
In table 7 and chapter 4 I do not understand why chitin from squid pens and fish gelatin is mixed with collagen. On the other hand why bovine derma is not included ? As far as I know it is un important source of collagen.
Authors response
Thanks for pointing out. It was mistakenly added to the collagen section. Now it has been removed from Table 7 and chapter 14.

Reviewer 2 Report
Title: Developments for collagen hydrolysate in biological, biochemical, and biomedical domains: A comprehensive review.
This manuscript aims at reviewing collagen hydrosilates at different levels or categories namely, processing technologies, targeted receptors, diseases, type of collagen and species which are divided by other subcategories. The authors highlight a novel way to hierarchically order and present these categories to provide information related to the scientific work of collagen as the main feature of the manuscript. The manuscript deals with a research topic that has been widely revised already in the literature. The novelty is not clear, moreover, in my opinion, the order proposed of the different sections and figures presented are confusing and difficult to follow. For instance, “process” is presented before that “species” or “type”, which is odd since processing is strongly dependent on the latter. The scope of the work is too wide, and in my opinion, it should be focused on those novel categories that are less researched but still relevant. Another source of confusion is that the authors refer to processed collagen as peptides, hydrosilates or gelatin, which is technically inappropriate. Gelatin results from partial hydrolysis of collagen, alfa and beta chains cannot be called peptides. Major revisions are needed for publication Other comments are presented below - The writing requires a major revision of the language (maybe by a native English speaker) - Table 1 is titled acidic hydrolysis and acid pre-treatment but reports the use of enzymes as well. - Table 4 includes whey protein as a source. Is this correct? - This review should also report the physical-chemical properties of collagen/gelatin reported (molecular weight, viscosity, Bloom, z-potential, etc.) - I think the review tries to cover too much information, and in this way, it generates more confusion than providing novel information. The subtitle Level 3.2.2.: Category 2: Novel method is confusing. The diagram (figure 1) is not clear to interpret and it's very confusing. I suggest perhaps focusing on manufacturing methods and different effects on health (diseases and the mechanism by which they act on them) - It is rare to talk about "receptors", especially I do not understand why this word is in quotes. If I understand correctly, they are trying to talk about the activation of different cell signalling pathways associated with hydrolyzed collagen effects (line 77). It conveys that the authors are not familiar with the subject (maybe some guidance is required by someone from cell biology). - The statement “The anti-metabolic syndrome peptides result in high bioactive functionalities through the use of invitro and Insilco approaches” (Lines 551-593) is confusing. - Also, a section in the manuscript reporting information about the inmuno response to collagens/gelatins/peptides should be included.
Author Response
Authors response
The authors are grateful for the valuable efforts of the respected reviewer. We have tried to cover most of the points, major and minor, in detail. Kindly overview the below-taken actions,
This manuscript aims at reviewing collagen hydrosilates at different levels or categories namely, processing technologies, targeted receptors, diseases, type of collagen and species which are divided by other subcategories. The authors highlight a novel way to hierarchically order and present these categories to provide information related to the scientific work of collagen as the main feature of the manuscript.
The manuscript deals with a research topic that has been widely revised already in the literature. The novelty is not clear,
The novelty is explained in two paragraphs (line 71 to 94) of the introduction along with the comparison of past reviews (lines 58 to 70). However, we highly rate the reservations of the respected reviewer. Therefore, we have further tried to explain the novelty in the following paragraphs,
The manuscript is particularly written to highlight the recent developments, especially the ones that are reported in the last 3-4 years, along with providing a basic knowledge regarding old techniques and developments. Meanwhile, it has an aim to highlight the unexplored areas related to the collagen.
As noted rightly by the respected reviewer that there is already available literature on collagen. In this regard, the recent reviews performed on collagen provide a holistic point of view about a single or a few of key aspect/s, that we have tried to highlight in the introduction (line 58 to 70) as a research gap. The limitation in the content of those reported reviews (line 58 to 70) has helped us to improvise the way of presenting the literature, along with the addition of the last 3 years’ literature particularly. In this regard, none of the recent reviews provides a near-to-complete hierarchical background that can be used as a guideline to a) understand the collagen literature (not just collagen by definition) in less time by research point of view, and to b) select a particular novel domain in collagen literature to commence future research. Both of these goals are successfully accomplished with the help of a proposed novel hierarchy. In short, the novel hierarchy purposefully provides a simple but detailed outlook of the decades of research on collagen, particularly of the last three to four years. Therefore, our prime novelty is the hierarchical structuring of the literature on collagen.
Additionally, each aspect (species, type, receptors, disease, process) is further presented with the relevant biological, biomedical, and biochemical applications. Just to clear one probable confusion, this review is not organised with respect to the applications (relevant biological, biomedical, and biochemical). Rather this review article is particularly organised to independently highlight the research on five highly researched aspects related to collagen intake. Therefore, each aspect (species, type, receptors, disease, process) is covered in detail regardless of having multiple applications in different publications. For example, Type is reported with all three kinds of applications (biological, biomedical, and biochemical).
Furthermore, based on the literature, the review adds the receptors or signalling pathways as one of the main domain for the research in collagen. Though the receptors or signalling pathways are proficiently covered in the literature, however, the scope is limited to only one or few research domains. In short, the available literature on signalling pathways, associated with collagen intake, provides a holistic overview that lacks the ease of selection for a particular domain for future research. In this review, different receptors or signalling pathways are individually covered in a systematic manner. This approach basically shortlists a few of many researched signalling pathways for the reader.
However, we are happy to modify our draft as per valuable suggestions of the respectable reviewer, if our answer is not found satisfactory.
moreover, in my opinion, the order proposed of the different sections and figures presented are confusing and difficult to follow. For instance, “process” is presented before that “species” or “type”, which is odd since processing is strongly dependent on the latter.
We acknowledge the proposed sequence by the respected reviewer and we have changed the sequence as per suggestion.
It is requested to kindly consider our point of view and allow us to keep the old sequence, if our below-mentioned thoughts are found logical,
The link between each of the five individual sections (categories) is intentionally not made at any point in the literature. As our main goal is to maintain the essence of hierarchy, which in some parts can be covered in chronological order. To the best of our knowledge, the earliest research on collagen was not covered in the following sequence: species, types, process. In fact, the early research papers covered “processes” as the highlighted novelty to process a particular species for extracting a particular type of collagen. This means that the researchers concentrate more on “process and species simultaneously” to achieve a particular type of collagen as a result. This is our basic thought process behind the proposed sequence.
However, we are willing to change the sequence as per suggestions in the next round if our answer is not found satisfactory.
Another source of confusion is that the authors refer to processed collagen as peptides, hydrosilates or gelatin, which is technically inappropriate. Gelatin results from partial hydrolysis of collagen, alfa and beta chains cannot be called peptides.
This was our mistake. We have now corrected the selected sections in the introduction (line 54 to 57).
Major revisions are needed for publication Other comments are presented below
Table 1 is titled acidic hydrolysis and acid pre-treatment but reports the use of enzymes as well.
The caption was wrong. Thanks for pointing it out. We have now corrected the caption as per the content. The new caption is as follow,
“ Table 1. Acid- and enzyme-based hydrolysis with common process parameters.”
- Table 4 includes whey protein as a source. Is this correct? -
It was a mistake. The complete set of materials are mistakenly missed in Table 4. The source was mixed bovine gelatine and whey protein. It is now corrected in Table 4 and the corresponding main text.
- I think the review tries to cover too much information, and in this way, it generates more confusion than providing novel information.
Thanks for pointing out one of the main weaknesses in our draft. To overcome these, we have taken multiple steps,
- Sections 2.6.1 to 2.6.4 and 2.6.5 to 2.6.6 are now merged into two independent categories (2.6.1 and 2.6.2). In this regard, the minor rephrasing is performed with proper grammar for new subchapters. The changes can be noted in lines 268 and 296. This will help to decrease the confusion for the reader.
- The conclusion is now rewritten in a concise but meaningful manner.
- A separate section is now added for future perspective and challenges. This is particularly made to highlight the novel areas separately, so the overall contribution of the review is not misleading and confused with the other information.
The subtitle Level 3.2.2.: Category 2: Novel method is confusing.
Thanks for pointing out this mistake.
The redundant words like “level” and “category” are now removed. Furthermore in the subheadings and Figures 3,4 and 6, the word “level” is also removed.
The diagram (figure 1) is not clear to interpret and it's very confusing. I suggest perhaps focusing on manufacturing methods and different effects on health (diseases and the mechanism by which they act on them) -
The extra information in Figure 1 b around the Main hierarchy is now presented in the same colour as the one in the main hierarchy. This helps to identify particular information easily and hence it reduces confusion. In this way, we are able to focus on various points in a single diagram.
However, we are willing to further change as per future instructions.
It is rare to talk about "receptors", especially I do not understand why this word is in quotes. If I understand correctly, they are trying to talk about the activation of different cell signalling pathways associated with hydrolyzed collagen effects (line 77). It conveys that the authors are not familiar with the subject (maybe some guidance is required by someone from cell biology).
It is rightly noted. We have now removed the double quotes. As per the literature, collagen intake in form of hydrolysates or gelatine has been reported to activate a particular neuronal signalling pathway. That results in a stimulus that cures the corresponding abnormality or disease.
However, we are happy to modify as per suggestions.
- The statement “The anti-metabolic syndrome peptides result in high bioactive functionalities through the use of invitro and Insilco approaches” (Lines 551-593) is confusing.
The line is now rephrased.
- Also, a section in the manuscript reporting information about the inmuno response to collagens/gelatins/peptides should be included.
Because the immunological benefits of collagen are associated with three aspects of novel hierarchy: 1) Type, 2) disease, and 3) receptors or signalling pathways. I.e., a separate section was not made to avoid any confusion. References 79, 82, 158, 165, and 168 are the relevant citations associated with immunological response caused by particular type of collagen intake. The above-mentioned references are explained in different sections as per their relevant appearance.
However, we highly respect the valuable comments of the reviewer. We will surely make a separate section if the reviewer will not find our reason justified.
Reviewer 3 Report
In the manuscript, the authors comprehensively summarized a wild range of research themes related to collagen hydrolysates. After reviewing this manuscript, it is recommended to consider this article for publication in the “Materials” before addressing the following issues:
- The structure of the review article is not summarized according to the title the authors set. What’s biological, biochemical, and biomedicine point-of-view on collagen hydrolysates should be involved in the manuscript.
- A detailed description of figure 2 should be involved in the manuscript.
- The captions of Figure 7 (A) to (D) are missing.
- Some abbreviations should be defined at the first time they showed in the manuscript. For instance, HP, UV, NG,
- The captions of tables should be more specific. For example, the caption of Table 5 is “Literature for Type I of collagen or collagen hydrolysate or gelatin”. Do the authors attempt to list all reports related to type I collagen?
- The authors should pay attention to the consistency of each unit for values in the manuscript.
- The authors may consider removing the term, “Level”, and merging 2.6.1-4 and 2.6.5-6 into two independent themes with title numbers of 2.6.1 and .2.6.2, respectively.
- It is recommended to alter the sequence of figure 4 and the main text of section 3.
- The category of “TYPE” showed in table 5 can be removed since the authors only summarized researches related to type I collagen, and this was described in the caption of Table 5.
- An additional section of “Perspectives” combined with table 9 is highly encouraged to be involved in the manuscript.
- Line 410: Type I -> Type II
Author Response
We are grateful to the reviewer for dedicated efforts that undoubtedly help us to improve our work. We have addressed all comments suggested by the Reviewer. Kindly find the details below,
In the manuscript, the authors comprehensively summarized a wild range of research themes related to collagen hydrolysates. After reviewing this manuscript, it is recommended to consider this article for publication in the “Materials” before addressing the following issues:
- The structure of the review article is not summarized according to the title the authors set. What’s biological, biochemical, and biomedicine point-of-view on collagen hydrolysates should be involved in the manuscript.
This point is highlighted at various places as per suggestion,
First in the main text, the following lines are added to highlight the type of corresponding functionality (biological, biochemical, and biomedical) resulted by collagen hydrolysate.
Line 848-850: Asai et al. [160] reports the biomedical functionality of Prolyl-hydroxyproline (Pro-Hyp) collagen through activation of P75 receptors.
Line 861: Zhu et al. [93] biologically activates the PI3K/AKT pathway by low molecular weight tripeptides of porcine collagen that cause downregulation of PTEN and upregulation of p-Akt, cyclin dependent protein kinase (CDK-2, CDK-4).
Line 870: Mei et al. [29] biomedically cures the wound through activation of PRRs (NOD2) and BD14 using marine extracted collagen.
Line 874-875: The research also reports decreases in biological expression levels of skin oxidation and matrix.
Line 889-890: Furthermore, the Table 8 has an individual column for highlighting the “biomedical” effects of the recent research. This Table is kept unchanged as it fulfils the suggested requirements for the topic.
- A detailed description of figure 2 should be involved in the manuscript.
Line 109 to 122: The description of the Figure 2 is added as per advise.
- The captions of Figure 7 (A) to (D) are missing.
Line 910-913: The caption is added. Furthermore, the citation in main text of each of Figure 7 (a) to (d) is also now provided.
- Some abbreviations should be defined at the first time they showed in the manuscript. For instance, HP, UV, NG,
Line 340, 392, 467 and 526: The “UV” is replaced with full form “Ultraviolet”.
Line 186, Table 2: The “HP” is replaced with full form “hydroxyproline”.
Tables 2, 3, 6, and 7: The word “NG” is replaced with full form “Not given”.
- The captions of tables should be more specific. For example, the caption of Table 5 is “Literature for Type I of collagen or collagen hydrolysate or gelatin”. Do the authors attempt to list all reports related to type I collagen?
The captions of Table 5 and 6 are now changed as following,
“Table 5. Examples for Type I collagen or collagen hydrolysate.”
“Table 6. Examples for Type II collagen or collagen hydrolysate.”
- The authors may consider removing the term, “Level”, and merging 2.6.1-4 and 2.6.5-6 into two independent themes with title numbers of 2.6.1 and .2.6.2, respectively.
We have taken corrective actions and removed the word “level” from all subchapters. Furthermore, the key advice of merging the 2.6.1-4 and 2.6.5-6 into two independent categories (2.6.1 and 2.6.2) are also done. In this regard, the minor rephrasing is performed with proper grammar for new subchapters. The changes can be noted at lines 268 and 296.
- It is recommended to alter the sequence of figure 4 and the main text of section 3.
The sequence is now altered to: first comes the text and then the Figure 4.
- The category of “TYPE” showed in table 5 can be removed since the authors only summarized researches related to type I collagen, and this was described in the caption of Table 5.
The corrective action is taken as per suggestion. The corresponding column in the Tables 5 and 6 are deleted.
- An additional section of “Perspectives” combined with table 9 is highly encouraged to be involved in the manuscript.
A separate section is now added to highlight the future perspective and challenges.
- Line 410: Type I -> Type II
Thanks for pointing out. The Corrective action is taken.

Reviewer 4 Report
In this manuscript, the authors present a complete review about collagen hydrolysate based on the literature of the last three decades. The review appears to be transversal exploring different aspects related to the hydrolysate collagen from the process to the disease passing by the various sources. The manuscript is well written and only few typos are present along all the manuscript even if a moderate English editing is needed to improve the quality of the manuscript. In addition, the graphical part is very accurate and only minor revision need to be done. However, what does not convince at all is the organization of the manuscript sometimes result confusing and the final goal of the review is not very clear especially in the conclusion part. The authors provide a lot of information about hydrolysate collagen but in my opinion, different organization of the paragraph should be adopted. Moreover, I would dedicate one paragraph about the application of hydrolysate collagen in regenerative medicine due to the fundamental role that it plays in this field and another paragraph to the future challenges. I would reconsider the publication of the present manuscript in Materials journal after some major revision as aforementioned. Here I report also some minor revision that have to be done:
- In the introduction at pp 2 line 54 the author define gelatin as the denatured form of the hydrolysate. However this definition is incorrect because Collagen hydrolysate is a polypeptide composite made by further hydrolysis of denatured collagen as reported by “Zhang, Z., Li, G., & Shi, B. I. (2006). Physicochemical properties of collagen, gelatin and collagen hydrolysate derived from bovine limed split wastes. Journal-society of leather technologists and chemists, 90(1), 23”. Please correct the definition;
- Figure 1 appear to be looks confused and there are too many information in it. Please adjust the figure making it more comprehensible;
- At pp 6 line 156 please correct the formula NAOH and the subscript of both Ca(OH)2;
- At pp 99 I don’t understand why the authors put “level” in the title of each sub-paragraph;
- In figure 3, figure 4 and figure 6 please remove the label with the hierarchical level because it is already clear.
- If the authors would like to explore the development of collagen hydrolysate in biomedical domain, why they did not deal with the possible application in tissue engineering?
- The conclusions paragraph appear to be confused. In fact, the authors give an overview of the article without come to any conclusions. Please rewrite this paragraph putting more effort about the conclusion to which the review arrive.
- Table 9 has no sense, please report this into paragraph called “future challenge”.
Author Response
We are grateful to the reviewer for dedicated efforts that undoubtedly help us to improve our work. We have addressed all comments suggested by the Reviewer. Kindly find the details below,
In this manuscript, the authors present a complete review about collagen hydrolysate based on the literature of the last three decades. The review appears to be transversal exploring different aspects related to the hydrolysate collagen from the process to the disease passing by the various sources. The manuscript is well written and only few typos are present along all the manuscript even if a moderate English editing is needed to improve the quality of the manuscript. In addition, the graphical part is very accurate and only minor revision need to be done. However, what does not convince at all is the organization of the manuscript sometimes result confusing and the final goal of the review is not very clear especially in the conclusion part. The authors provide a lot of information about hydrolysate collagen but in my opinion, different organization of the paragraph should be adopted.
Authors response
In this regard, extensive and meaningful modifications are made in the main draft,
The structure of the review article is now summarized according to the title the authors set. What’s biological, biochemical, and biomedicine point-of-view on collagen hydrolysates should be involved in the manuscript.
The following lines are added to highlight the type of corresponding functionality (biological, biochemical, and biomedical) resulted by collagen hydrolysate.
Line 848-850
Asai et al. [160] reports the biomedical functionality of Prolyl-hydroxyproline (Pro-Hyp) collagen through activation of P75 receptors.
Line 861
Zhu et al. [93] biologically activates the PI3K/AKT pathway by low molecular weight tripeptides of porcine collagen that cause downregulation of PTEN and upregulation of p-Akt, cyclin dependent protein kinase (CDK-2, CDK-4).
Line 870
Mei et al. [29] biomedically cures the wound through activation of PRRs (NOD2) and BD14 using marine extracted collagen.
Line 874-875
The research also reports decreases in biological expression levels of skin oxidation and matrix.
Line 889-890
Furthermore, the Table 8 has an individual column for highlighting the “biomedical” effects of the recent research. This Table is kept unchanged as it fulfils the suggested requirements for the topic.
Page 4, Line 108 to 121 Figure 2 is now explained, which was missing earlier. This has help to organize the text better than before.
Page 9 and 10, Line 276 to 327 The whole section of hydrostatic extraction is now simplified. The redundant information is now removed, and the section is now comprised of just two (2) subsection instead of six (6). This has help to organize the text better than before.
Page 8, 13 and 24 The redundant words like “level” in Figures are now removed in Figures 3,4 an 6. Furthermore, Figure 7 caption is now added with detail.
I would reconsider the publication of the present manuscript in Materials journal after some major revision as aforementioned. Here I report also some minor revision that have to be done:
- In the introduction at pp 2 line 54 the author define gelatin as the denatured form of the hydrolysate. However this definition is incorrect because Collagen hydrolysate is a polypeptide composite made by further hydrolysis of denatured collagen as reported by “Zhang, Z., Li, G., & Shi, B. I. (2006). Physicochemical properties of collagen, gelatin and collagen hydrolysate derived from bovine limed split wastes. Journal-society of leather technologists and chemists, 90(1), 23”. Please correct the definition;
Authors response
The definition is now corrected with the suggested references citation.
- Figure 1 appear to be looks confused and there are too many information in it. Please adjust the figure making it more comprehensible;
Authors response
The extra information in Figure 1 b around the Main hierarchy is now presented in same colour as like the one in main hierarchy. This helps to identify a particular information easily and hence it reduces the confusion.
- At pp 6 line 156 please correct the formula NAOH and the subscript of both Ca(OH)2;
Authors response
The formulae for both NaOH and Ca(OH)2 is corrected. Furthermore, the corresponding formulae in Table 2 is also corrected.
- At pp 99 I don’t understand why the authors put “level” in the title of each sub-paragraph;
Authors response
The “word” level is now removed from all headings and sub-headings.
- In figure 3, figure 4 and figure 6 please remove the label with the hierarchical level because it is already clear.
Authors response
The label “level” is now removed from Figures 3, 4 and 6.
- If the authors would like to explore the development of collagen hydrolysate in biomedical domain, why they did not deal with the possible application in tissue engineering?
Authors response
The applications in tissue engineering is mentioned at various sections in the draft now,
Line 362
Line 428
Line 697
Line 758
- The conclusions paragraph appear to be confused. In fact, the authors give an overview of the article without come to any conclusions. Please rewrite this paragraph putting more effort about the conclusion to which the review arrive.
Authors response
The conclusion is now rewritten as per suggestions.
- Table 9 has no sense, please report this into paragraph called “future challenge”.
Authors response
A separate section is now added to highlight the future challenges.

Round 2
Reviewer 2 Report
Title: Developments for collagen hydrolysate in biological, biochemical, and biomedical domains: A comprehensive review The authors have considered most of the recommendations and the manuscript have been improved. I agree that the inclusion of the section at the end; Future Perspective and challenges help to highlight the contribution of the work. However there still some improvements needed 1. If the order of the information presented in the original version of the manuscript is to be kept, as requested by the authors, the description of the rationale behind the new hierarchically (levels) proposed to organize the information from the literature on collagen should be more clearly presented. For instance from the source, type, processing, characteristics application, etc., typical reporting order. 2. Check the numbering of some headings in the text (subheadings 4 after heading 2) 3. Include a section summarizing information regarding immune responses to collagens/gelatins/peptides.
Author Response
We highly appreciate the valuable feedback and time of the reviewer.
We have taken the following actions regarding the revisions in round 2,
- The headings numbers are now corrected.
- The preferred orientation of the respected reviewer is kept the same.
- A short summary of the immune response of collagen intake is added (line 907 to 914).
We are happy to further incorporate changes as it will help us to improve the quality of the manuscript for the readers.
Regards,

Reviewer 3 Report
The authors have addressed all issues the reviewer had delivered, and the quality of this manuscript has been significantly improved. Thus, it is believed that this article is ready for publication in "Materials".
Author Response

(The authors gave the same response as above.)

Reviewer 4 Report
The authors reply to all the comment of the reviewer and extensive modifications have been made increasing the quality of the manuscript. Now the manuscript organization is more logical and also the scientific soundness of the manuscript have been increased. I appreciate also the paragraph 2 with the division between species as collagen sources and species as collagen subject. Figure 2 appear more clear now that the authors used different colours for the different hierarchical level. The conclusion part now reflect the topic addressed in the paper and also the future perspective paragraph perfectly fit with table 9. I would suggest the publication of the present manuscript in material journal after the extensive and meaningful modification that the authors have made.
Author Response
We highly appreciate the valuable feedback and time of the reviewer.
We have taken the following actions regarding the revisions in round 2,
- The headings numbers are now corrected.
- A short summary of the immune response of collagen intake is added (line 907 to 914).
We are happy to further incorporate changes as it will help us to improve the quality of the manuscript for the readers.
Regards,